# On global convergence of ResNets: From finite to infinite width using linear parameterization

**Raphaël Barboni**
ENS - PSL Univ.
`raphael.barboni@ens.fr`

**Gabriel Peyré**
CNRS and ENS - PSL Univ.
`gabriel.peyre@ens.fr`

**François-Xavier Vialard**
LIGM, Univ. Gustave Eiffel, CNRS
`francois-xavier.vialard@u-pem.fr`

## Abstract

Overparameterization is a key factor in the absence of convexity to explain global convergence of gradient descent (GD) for neural networks. Beside the well studied lazy regime, infinite width (mean field) analysis has been developed for shallow networks, using convex optimization techniques. To bridge the gap between the lazy and mean field regimes, we study Residual Networks (ResNets) in which the residual block has linear parameterization while still being nonlinear. Such ResNets admit both infinite depth and width limits, encoding residual blocks in a Reproducing Kernel Hilbert Space (RKHS). In this limit, we prove a local Polyak-Lojasiewicz inequality. Thus, every critical point is a global minimizer and a local convergence result of GD holds, retrieving the lazy regime. In contrast with other mean-field studies, it applies to both parametric and non-parametric cases under an expressivity condition on the residuals. Our analysis leads to a practical and quantified recipe: starting from a universal RKHS, Random Fourier Features are applied to obtain a finite dimensional parameterization satisfying with high-probability our expressivity condition.

## 1 Introduction

State of the art supervised learning methods are based on deep neural networks, sometimes heavily overparameterized, which perfectly fit training data or even noisy data while exhibiting good generalization properties. Such a behaviour appears as a paradox and questions the established theory of "bias-variance trade-off" [9]. That an overparameterized model can fit data perfectly comes as no surprise but this capability does not explain the observed generalization properties. Towards a better understanding of it, one first needs to understand the optimization procedure in the parameter space that selects the interpolation map. This question is tightly linked with the parameterization of the space of maps that are explored and state of the art parameterizations have emerged in the past years. One key architecture that is ubiquitous in deep learning are skip connections, heavily used in *Residual Neural Networks* (ResNets) [25] and it has led to state of the art results in supervised learning. ResNets actually allow one to consider a very large number of layers [59].

**Continuous models.** Passing to the limit of infinite depth allows the connection with continuous models (Neural ODE) for which theoretical methods and new algorithms can be designed [11, 56]. Indeed, the similarities between ResNet architectures and discrete numerical schemes motivated the introduction of a continuous neural ODE

$$\dot{z}_t = v(W_t, z_t) \quad \forall t \in [0, 1], \tag{1}$$

36th Conference on Neural Information Processing Systems (NeurIPS 2022).

where $W \in L^2([0, 1], \mathbb{R}^m)$ is the parameter of the model and $v : \mathbb{R}^m \times \mathbb{R}^q \to \mathbb{R}^q$ is a *residual transformation* whose output is the *residual term*. These models correspond to limiting models of a discrete ResNet whose depth $D$ tends to infinity. Therefore, their study brings a theoretical framework for understanding deep ResNet architectures, and more generally very deep NNs [19, 20]. Moreover, their mathematical analysis is facilitated since it allows one to leverage a large body of works and tools from analysis and in particular the theory of optimal control [47]. Conversely, methods from numerical analysis can bring inspiration for designing new architectures and new optimization algorithms [39].

**RKHS parameterization.** Most often in the literature studying the training properties of ResNets, the considered residual transformations are *Multi-Layer Perceptrons (MLP)* [16, 2, 24]. Those consist in the composition of several trained linear layers alternatively composed with a non-linear activation function. A 2-layer MLP with width $r$ reads:

$$v : ((W, U), z) \mapsto W\sigma(Uz), \tag{2}$$

where $U \in \mathbb{R}^{r \times q}$ and $W \in \mathbb{R}^{q \times r}$ are the parameters for the "hidden" and the "visible" layer respectively and $\sigma : \mathbb{R} \to \mathbb{R}$ is a non-linear *activation function* applied component-wise. Popular activation functions are for example the ReLU or the Swish function. Provided with these activations, MLPs enjoy a nice universal approximation property as shown in the seminal work of Barron [6].

In contrast, we consider here a setting where the residual term is linear w.r.t. the parameters while still being nonlinear w.r.t the inputs. Given a feature map $\varphi : \mathbb{R}^q \to \mathbb{R}^r$, we consider as space of residuals the space:

$$V := \{v : z \mapsto W\varphi(z) | W \in \mathbb{R}^{q \times r}\}, \tag{3}$$

where the matrices $W \in \mathbb{R}^{q \times r}$ are the trained parameters. Compared to Eq. (2), this can be seen as an MLP where the hidden layer is fixed by introducing the feature map $\varphi : z \mapsto \sigma(Uz)$ for some feature matrix $U$. As is standard, the gradient of some loss $L$ w.r.t. $W$ is computed in the sense of the Frobenius metric on the set of matrices:

$$\forall W, W' \in \mathbb{R}^{q \times r}, \ \langle W, W' \rangle = \mathrm{Tr}(W^\top W'). \tag{4}$$

Such an $L^2$ penalization induces a metric structure on $V$ through the identification $v \leftrightarrow W$ in Eq. (3):

$$\forall v, v' \in V, \ \langle v, v' \rangle_V := \langle W, W' \rangle. \tag{5}$$

As a finite dimensional space of continuous maps, $V$ has the structure of *Reproducing Kernel Hilbert Space* (RKHS). Moreover, as pointed out in [5], the space $V$ has a natural infinite width limit or mean field limit which is an infinite dimensional RKHS.

In this paper, we are interested in understanding the convergence properties of Gradient Descent (GD) on a ResNet model for which the residual layers are encoded in a – possibly infinite-dimensional – vector-valued RKHS $V$. For $V$ as in Eq. (3), we stress out that, as the metric on $V$ is induced by the one on $\mathbb{R}^{q \times r}$, GD on $V$ for this metric is strictly equivalent to GD on $\mathbb{R}^{q \times r}$ with the Frobenius metric. Our model is defined as follows:

**Definition 1** (RKHS Neural ODE (RKHS-NODE)). *Let $V$ be a RKHS of vector-fields over $\mathbb{R}^q$ and $A \in \mathbb{R}^{q \times d}$, $B \in \mathbb{R}^{d' \times q}$. Then for $v \in L^2([0, 1], V)$ and a data input $x \in \mathbb{R}^d$, the RKHS-NODE's output is $F(v, x) := Bz_1$, where $z$ is the solution to the* forward problem

$$\dot{z}_t = v_t(z_t) \quad and \quad z_0 = Ax. \tag{6}$$

*The variable $v$ will thereafter be called* control parameter.

**Remark 1.** *Note that the matrices $A$ and $B$ are fixed and only the control parameter $v$ is trained. However, we argue that our approach can be simply adapted to the case where $B$ is trained, following for example the proof of [43]. Training $A$ seems more challenging as the model is highly non-linear w.r.t. this parameter.*

**Relevance of the RKHS model.** The main difference between the model of Definition 1 and standard ResNets is linearity in the parameters of the residual blocks. As a comparison, a 2-layer MLP is nonlinear w.r.t. the parameters of the hidden layers. However, this linearity assumption does not impact the expressivity of the model, but only its training dynamic. **(i)** Indeed, considering $V$ to be a Random Features approximation (c.f. Eq. (20)) of some universal RKHS, the residual blocks

are as expressive as a 2-layer MLP since both are dense in the space of continuous functions. **(ii)** Up to the cost of adding a supplementary variable, the dynamical system parameterized by a 2-layer MLP can be expressed as a model which is linear w.r.t. its parameters [56, Section 3.2]. Only the training dynamic between these two architectures differs. Also, this assumption of linearity in the parameters also prevents the use of normalization layers. In this direction, [61] has shown that ResNets without normalization but proper initialization of the weights can lead to robust training and similar performance on the train set than standard ResNets. Finally, the model of Definition 1 still retains the effect of depth and the nonlinearity w.r.t. the input. Due to composition of these residual blocks the model's output is still highly non-linear w.r.t. parameters. For these reasons, we consider this model as an important step towards the study of the general case.

In turn, this linearity in parameters naturally leads to an RKHS parameterization which has two important benefits on the theoretical side: **(i)** Flows of vector-fields as implemented by our model in Eq. (6) have already been studied theoretically and for applications in image registration problems [58, 8, 44]. Under some regularity assumptions on the considered RKHS $V$, one can show that the model's output corresponds to the invertible action of a diffeomorphism by composition on the input [55]. This property was already used in [51] to implement models of *Normalizing Flows* [29] with applications in generative modeling. **(ii)** There is an important literature in Machine Learning about Kernel methods [52]. In practice, various sub-sampling methods exist in order to approximate infinite-dimensional RKHSs with finite-dimensional spaces generated by *Random Fourier Features* (RFF) [48, 49]. Thereby, leveraging results on the approximation bound for RFF [54, 53], we show that the expressiveness properties of universal kernels, such as the Gaussian kernel, can be efficiently recovered using residuals of the form Eq. (3) with a finite number of neurons.

To further support the practical applicability and the relevance of our model in comparison with standard architectures, we report in the supplementary material (Appendix A) numerical experiments on MNIST and CIFAR10 datasets. They show that – as predicted by our theory – our model can be trained in these cases to almost zero loss. But more importantly, they show that our model is able to generalize well on the test dataset with performances that are similar to those of classical ResNets.

**Supervised learning.** We consider a map $F$ from $\mathcal{H} \times \mathbb{R}^d$ to $\mathbb{R}^{d'}$ for some Hilbert space of parameter $\mathcal{H}$ (e.g. the model of Definition 1 with $\mathcal{H} = L^2([0,1], V)$) and a training dataset consisting on a family of inputs $(x^i)_{1 \leq i \leq N} \in (\mathbb{R}^d)^N$ and target outputs $(y^i)_{1 \leq i \leq N} \in (\mathbb{R}^{d'})^N$. Then for every parameter $v \in \mathcal{H}$, we define the associated *Empirical Risk* as:

$$L(v) := \frac{1}{2N} \sum_{1 \leq i \leq N} \|F(v, x^i) - y^i\|^2. \tag{7}$$

**Remark 2.** *For simplicity we consider here the Euclidean square distance as a loss on the output space $\mathbb{R}^{d'}$, but our results generalize to any smooth loss satisfying a Polyak-Lojasiewicz inequality (c.f.[10]), e.g. any smooth strongly convex loss.*

Training the model $F$ then amounts to finding a parameter $v^* \in \arg\min_{v \in \mathcal{H}} L(v)$. In order to perform such an *empirical risk minimization (ERM)* task we consider GD on $v$. For a small step size $\eta$, for some initialization $v^0 \in \mathcal{H}$ and for every discrete time step $k \in \mathbb{N}$, the training dynamic reads:

$$v^{k+1} = v^k - \eta \nabla L(v^k).$$

Note that we do not consider any additional regularizing term on the loss. In classical supervised learning one would seek for a minimizer of the "regularized" loss $L(v) + \lambda \mathcal{R}(v)$, with $\lambda > 0$ a constant and $\mathcal{R}$ a coercive regularization function. We are here interested in the non regularized setting, i.e. $\lambda = 0$, often used in practice. In this case, the generalization property of the computed map is argued to potentially come from the optimization method that shall select an adequate minimizer of the loss. This implicit regularization depends on the choice of the optimization method [42].

## 2   Related works and contributions

Recently, several works have addressed the problem of proving convergence of (stochastic) GD in the training of NNs. If the convergence properties of GD are well understood for NNs that are linear w.r.t. input [24, 7, 64], it is not the case for non-linear NNs. In [34, 33, 17], the authors focus on the training of "shallow" two layers fully connected NNs and establish convergence of GD in an

overparameterized setting where width of the intermediary layer scales polynomially with the size $N$ of the dataset. More recently, with the same setup, [62] showed that the neurons of a teacher network are recovered by a student network optimized with GD as long as the width of the student network is higher than the teacher's one. Formally, their analysis is similar to ours as the result holds if the loss at initialization is already sufficiently low and the proof relies on Polyak-Lojasiewicz inequalities verified by the loss landscape.

**Infinite depth.** The works of [16, 2, 64, 32, 63, 35, 12, 43] extend those results to arbitrary deep NN in the overparameterized setting. Specifically, the results in [16, 2, 35] apply to deep ResNets. The best result seems to be achieved in [43], with convergence as soon as the last layer has a width $m = \Omega(N^3)$ and at best with linear width. A common feature for those works is to rely on the fact that, for a sufficiently high number of parameters, the model can be well approximated by a linear model corresponding to its first order expansion around the initialization. In [15] this phenomenon, called "lazy regime", is attributed to an inappropriate scaling of the parameters. On the other hand, [36, 35] refer to this phenomenon as "linear" or "kernel regime" and relate it the constancy of the *Neural Tangent Kernel (NTK)* introduced in [26]. However, in all those works the width of intermediary layers has to depend on the depth $D$ of the network. Therefore, these results do not apply to the training of the model in Eq. (1), corresponding to the limit $D \to +\infty$.

**Infinite width.** The other direction of over-parameterization, analyzed in several works [41, 14, 40, 27, 38, 21, 46] is to consider the limit of infinitely wide layers. In such a "mean-field" setting, the model is parameterized by the distribution of the parameters at each layer. In [14, 41, 40, 27] the training dynamic is analyzed as a gradient flow in the Wasserstein space [3], showing that the only stationary distributions are global minimizers of the empirical risk. In [21] a similar result is showed for deep NN with an arbitrary number of infinitely wide layers. In [13, 1], local linear convergence towards the global optimum is shown for two layers NNs in a teacher-student setup with regularized loss. Finally, [38] analyzes the convergence of continuous ResNets with infinitely wide residual layers and shows that every critical point is a global minimizer of the empirical risk. We stress out that these results only apply to infinitely wide NNs. It is not clear if this mean-field limit extends to the parametric setting of MLPs with the Euclidean metric on their parameters. In contrast, a RKHS structure naturally arises when considering a linear parameterization of the residuals. Assumption 1 and Assumption 2 can be satisfied both in a parametric setting with a finite number of features and in a mean-field setting limit where the residuals are generated by a universal kernel.

**Contributions.** We show convergence results for GD in the training of RKHS-NODEs (see Definition 1). These correspond to infinitely deep continuous ResNets with linear parameterization of the residuals. Our first main contribution, in Section 4, shows that under some regularity and expressivity assumptions on the residuals, the associated empirical risk satisfies a (local) Polyak-Lojasiewicz Property 2. A consequence is Theorem 2, which states global convergence of GD towards a global optimum (zero training loss) under the condition that the loss at initialization is already sufficiently low. In the limit where the loss at initialization is arbitrarily small, we recover a linear regime as described in [36, 35]. Our second contribution, in Section 5, shows how this condition for global convergence can be enforced using suitably chosen first and last linear layers. Thereafter, we show how the assumptions of Theorem 2, can be satisfied for RKHSs generated by a finite number of Random Features, with high probability over the choice of these features. For any dataset $(x^i, y^i)_{1 \leq i \leq N} \in (\mathbb{R}^d \times \mathbb{R}^{d'})^N$, we conclude in Theorem 3 to convergence of GD towards a global minimum of Eq. (7) with high probability when the width of the layers scales polynomially w.r.t. the size of the dataset $N$ and the inverse input data separation $\delta^{-1}$.

Finally, we point out that some of our results can be seen as a generalization of existing results concerning convergence of GD for the training of linear NNs [24, 7, 64]. We explain in Appendix E how, following the line of our analysis, one can for example recover [64, Theorem 3.1.]. However, if Definition 1 encompasses linear ResNets as a special case, we stress that Theorem 2 applies to a way larger class of models.

**Notations.** In what follows $\|.\|$ denotes the Euclidean $\ell^2$ norm for vectors and the Frobenius norm for matrices. For matrices the spectral norm is denoted $\|.\|_2$, the smallest (resp. greatest) singular value is denoted $\sigma_{\min}$ (resp. $\sigma_{\max}$) and for symmetric matrices the smallest (resp. greatest) eigenvalue is denoted $\lambda_{\min}$ (resp. $\lambda_{\max}$). Given some Hilbert space $\mathcal{H}$, the functional Hilbert space $L^2([0,1], \mathcal{H})$ is denoted $L^2(\mathcal{H})$ or $L^2$ when there is no ambiguity. The notation $\mathcal{O}$ (resp. $\Omega$) means asymptotically inferior (resp. superior) up to multiplicative constant.

# 3 Analysis of convergence for overparameterized models

In this section, we review methods for analyzing the convergence of overparameterized machine learning models based on [36, 35]. We refer to Appendix B for detailed proofs of the statements.

As presented above, we consider an optimization over the variable $v$ in some Hilbert space $\mathcal{H}$, with fixed input and output data, say $v \mapsto F(v) := [F(v, x_i)]_{i=1,\dots,N}$. Therefore, the empirical risk is a function of the parameters $v \in \mathcal{H}$. We say that the model is *overparameterized* whenever the dimension $\dim(\mathcal{H})$ of the parameter space is much larger than the dimension of the output space of $F(v)$, here $d'N$. The RKHS-NODE model defined $F$ in Definition 1 falls into this category as $\mathcal{H}$ is the infinite dimensional functional space $L^2([0, 1], V)$.

## 3.1 A (local) Polyak-Lojasiewicz property

When dealing with overparameterized models, one cannot expect the loss to be convex but one expects the model to perfectly fit the data, that is to reach the global minimum value of $0$. In fact, for a sufficient number of parameters, the loss landscape typically possesses a continuum of infinitely many global minima and is non-convex in any neighbourhood of a global minima [36]. One thus rather needs to rely on a set of functional inequalities allowing to control the decrease rate of the loss along GD [37, 10].

**Definition 2** ((local) Polyak-Lojasiewicz property). *Let $L : \mathcal{H} \to \mathbb{R}_+$ be a differentiable function. We say that $L$ satisfies a (local) Polyak-Lojasiewicz (PL) property if there exist positive continuous functions $m, M : \mathbb{R}_+ \to \mathbb{R}_+^*$ s.t. for every $v \in \mathcal{H}$*

$$2m(\|v\|)L(v) \leq \|\nabla L(v)\|^2 \leq 2M(\|v\|)L(v). \tag{8}$$

Such functional inequalities have already shown to be relevant for proving convergence guarantees in the training of NNs [22]. A first consequence for a loss $L$ which satisfies the (local) PL property of Definition 2 is that it does not admit any spurious local minima but only global minima. Also, if the training dynamic is bounded, then $m$ and $M$ are uniformly lower- and upper-bounded along the dynamic, implying that $L$ decreases at a linear rate. In most cases, $m$ and $M$ are degenerate when $\|v\| \to +\infty$. When the dynamic is not bounded, $L$ can thus decrease to $0$ slower than at a linear rate or even converge towards a strictly positive limit.

## 3.2 Local convergence result

Because of the degeneracy of $m$ and $M$, it is in general not possible to conclude an unconditional convergence of GD towards a global minimizer of the empirical risk. However, PL inequalities are sufficient to prove convergence when the problem is not too hard to solve, that is when the loss at initialization is not too high. Moreover, when using gradient descent stepping, one needs to make a supplementary smoothness assumption on the empirical risk $L$. This ensures that the loss decreases at each gradient step for a sufficiently small step size.

**Definition 3** (Smoothness, Definition 2 of [36]). *Let $\beta \geq 0$ be a constant. We say that the function $L : \mathcal{H} \to \mathbb{R}$ is $\beta$-smooth if for every $v, v' \in \mathcal{H}$: $|L(v') - L(v) - \langle \nabla L(v), v' - v \rangle| \leq \frac{\beta}{2}\|v' - v\|^2$.*

The local PL property combined with this smoothness assumption then gives a local convergence result for the convergence of GD towards a global minimizer of the empirical risk.

**Theorem 1** (Theorem 6 of [36]). *Let $L : \mathcal{H} \to \mathbb{R}_+$ be a loss function satisfying a local PL property with local constants $m$ and $M$. Let $v^0 \in \mathcal{H}$ and $R \geq 0$ be such that*

$$2\sqrt{2}\frac{\sqrt{M(\|v^0\| + R)}}{m(\|v^0\| + R)}\sqrt{L(v^0)} \leq R. \tag{9}$$

*Furthermore, assume that $L$ is $\beta$-smooth within the ball $B(v^0, R)$. Then for a step size $\eta \leq \beta^{-1}$, GD with initialization $v^0$ and step size $\eta$ converges towards a global minimizer of $L$ with a linear convergence rate and inside a ball of radius $R$. More precisely, for every $k \geq 0$:*

$$L(v^k) \leq (1 - m(\|v^0\| + R)\eta)^k L(v^0) \quad and \quad \|v^k - v^0\| \leq R, \ \forall k \geq 0. \tag{10}$$

# 4    Properties of RKHS-NODE

In this section we analyze the convergence of GD in the training of the infinitely deep ResNet model of Definition 1. Note that such a model is overparameterized in depth as the parameter space is the infinite dimensional space $L^2([0,1],V)$ and overparameterization can also come from width when the RKHS is high (or even infinite) dimensional. Therefore, our proof of convergence heavily relies on a PL property verified by the empirical risk.

Recall that we consider the training of deep ResNets with a linear parameterization of the residuals. The set of residuals is as in Eq. (3) with the metric of Eq. (5) induced by the Frobenius metric (Eq. (4)). This provides $V$ with a RKHS structure [4], whose associated kernel is given for any $z, z' \in \mathbb{R}^q$ by $K(z,z') := \langle \varphi(z), \varphi(z') \rangle \operatorname{Id}_q$, and whose associated feature map is given by $\varphi$.

**Remark 3.** *The definition of $\langle .,. \rangle_V$ in Eq. (5) requires* $\operatorname{Span}(\varphi(\mathbb{R}^q)) = \mathbb{R}^r$ *to associate each $v \in V$ to a unique $W \in \mathbb{R}^{q \times r}$. This is satisfied by all the feature maps $\varphi$ we consider in the following.*

Given a training dataset composed of input data points $(x^i)_{1 \le i \le N} \in (\mathbb{R}^d)^N$ and of target data points $(y^i)_{1 \le i \le N} \in (\mathbb{R}^{d'})^N$ we are interested in the task of minimizing the empirical risk of Eq. (7) by GD over $v$. Analogously to back-propagation in discrete NNs architectures, the gradient of $L$ can be expressed thanks to a backward equation derived by adjoint sensitivity analysis [47].

**Property 1.** *Let $L$ be the empirical risk in Eq.* (7) *associated with the RKHS-NODE model with a quadratic loss. Let $K$ be the kernel function associated with the RKHS $V$. Then $L$ is differentiable on $L^2([0,1],V)$, with for every $v \in L^2([0,1],V)$, $\nabla L(v) = \sum_{i=1}^N K(.,z^i)p^i$, where for each index $i \in [\![1,N]\!]$, $z^i$ is the solution of Eq.* (6) *with initial condition $Ax^i$ and the* adjoint *variable $p^i$ is the solution to the* backward *problem:*

$$\dot{p}_t^i = -Dv_t(z_t^i)^\top p_t^i \quad and \quad p_1^i = -\frac{1}{N} B^\top (Bz_1^i - y^i). \tag{11}$$

## 4.1    PL property of RKHS-NODE

Following the line of proof sketched in Section 3, we show how to derive PL inequalities of the form Eq. (8) for the empirical loss associated with the RKHS-NODE model. For that purpose we make a few assumptions about the RKHS $V$. The first one concerns its regularity and allows us to control the solutions of Eqs. (6) and (11).

**Assumption 1** ((strong) Admissibility). *We say that the RKHS $V$ is* (strongly) admissible *if it is continuously embedded in $W^{2,\infty}(\mathbb{R}^q, \mathbb{R}^q)$. More precisely, there exists a constant $\kappa > 0$ s.t.*

$$\forall v \in V, \quad \|v\|_\infty + \|Dv\|_{2,\infty} + \|D^2v\|_{2,\infty} \le \kappa \|v\|_V. \tag{12}$$

Assuming $V$ is embedded in $W^{1,\infty}(\mathbb{R}^q, \mathbb{R}^q)$ is natural to ensure the regularity of the flow generated by the control parameter [55, 58] and suffices to prove convergence of a continuous gradient flow on the parameter $v$. Assumption 1 is a bit stronger because a supplementary smoothness result on the loss landscape is necessary to prove convergence of discrete GD (c.f. Definition 3). In practice, $\kappa$ can be computed for smooth kernels thanks to Property 4 in Appendix D. For example, the RKHS associated with the Gaussian kernel $k : r \mapsto e^{-r^2/2}$ is (strongly) admissible with $\kappa = 2 + \sqrt{3}$.

The second assumption is related to the expressiveness of $V$ and is a weaker form of the classical universality property of RKHSs.

**Assumption 2** ($N$-universality). *Let $K$ be the kernel function associated with the RKHS $V$. For a family of points $(z^i)_{1 \le i \le N} \in (\mathbb{R}^q)^N$, we define the associated kernel matrix as the block matrix $\mathbb{K}((z^i)_i) := (K(z^i, z^j))_{1 \le i,j \le N}$.*
*More precisely we assume for every $\delta > 0$:*

$$\Lambda := \sup_{(z^i) \in (\mathbb{R}^q)^N} \lambda_{\max}(\mathbb{K}((z^i)_i)) < +\infty \quad and \quad \lambda(\delta^{-1}) := \inf_{\substack{(z^i) \in (\mathbb{R}^q)^N \\ \min_{i \ne j} \|z^i - z^j\| \ge \delta}} \lambda_{\min}(\mathbb{K}((z^i)_i)) > 0. \tag{13}$$

Assumption 2 is required in order to ensure the expressivity of our model, quantified by the conditioning of the kernel matrix $\mathbb{K}$ and by $\Lambda$ and $\lambda$. The choice of the RKHS $V$ may thus have a

significant impact on training. In particular, satisfying Assumption 2 requires having $V$ of dimension $m \geq N$, but it can be satisfied for finite dimensional RKHSs of dimension $m \leq N^q$, for example by considering a polynomial kernel, or by RKHSs of dimension $m \geq poly(N, q)$ with high probability on the sampling of random features, as shown in Section 5. On the other hand, even though the existence of $\lambda$ follows from compactness arguments, it seems to be hardly analytically tractable even for classical kernels such as the Gaussian kernel. Therefore, if, in theory, prior knowledge of the data distribution might allow to optimize the choice of kernel, we expect the selection of an optimal kernel to be an intractable problem in practice. Instead, cross-validation techniques can be used to select a suitable kernel.

**Remark 4.** *For a RKHS $V$ as in Eq. (3), the properties of $V$ only depend on $\varphi$. An interesting example is when $\varphi : z \mapsto \sigma(Uz)$ with $\sigma$ an activation function applied component-wise and $U$ a fixed feature matrix. In Section 5 we show that, when considering the complex activation $\sigma : t \mapsto e^{-it}$, both assumptions can be satisfied with high probability. On the other hand, Assumption 1 is not satisfied when considering $\sigma = ReLU$ due to its non-smoothness at $0$.*

**Remark 5.** *Note that $\Lambda$ could also be allowed to depend on some parameters, such as $\max \|z^i\|$. However, as it is a more critical aspect of our analysis, we prefer to highlight the dependency of $\lambda$ w.r.t. $\min_{i \neq j} \|z^i - z^j\|$. For all the RKHSs studied here we always have $\Lambda \leq N$.*

The following PL property is satisfied by the risk $L$. Property 2 is proven in Appendix C.2.

**Property 2** (RKHS-NODE satisfy PL). *Assume $V$ satisfies Assumption 1 with $\kappa$ and Assumption 2 with $\lambda$ and $\Lambda$. Let $L$ be the empirical risk in Eq. (7) associated with the RKHS-NODE model of Definition 1. Then $L$ satisfies the PL inequalities of Definition 2 with $m$ and $M$ given by:*

$$M(R) = \frac{1}{N}\sigma_{\max}(B^\top)^2 \Lambda e^{2\kappa R}, \quad m(R) = \frac{1}{N}\sigma_{\min}(B^\top)^2 \lambda \left(\sigma_{\min}(A)^{-1}\delta^{-1}e^{\kappa R}\right) e^{-2\kappa R}, \quad (14)$$

*where $\delta := \min_{i \neq j} \|x^i - x^j\|$ is the data separation.*

*Sketch of proof.* Assumption 1 can be used to have estimates on the solutions $z^i$ of the forward problem Eq. (6) and on the solutions $p^i$ of backward problem Eq. (11). This gives for every indices $i, j \in [\![1, N]\!]$ and every $t \in [0, 1]$:

$$\|z_t^i - z_t^j\| \geq \sigma_{\min}(A)\|x^i - x^j\|e^{-\kappa\|v\|_{L^2}},$$

where $z^i$ solves Eq. (6) with initial condition $Ax^i$, and:

$$e^{-2\kappa\|v\|_{L^2}}\|p_1^i\|^2 \leq \|p_t^i\|^2 \leq e^{2\kappa\|v\|_{L^2}}\|p_1^i\|^2.$$

Moreover using the initial condition $p_1^i = -\frac{1}{N}B^\top(Bz_1^i - y^i)$ we have:

$$\frac{2\sigma_{\min}(B^\top)^2}{N}L(v) \leq \sum_{i=1}^N \|p_1^i\|^2 \leq \frac{2\sigma_{\max}(B^\top)^2}{N}L(v).$$

Then denoting $\tilde{p}_t$ the vector of stacked $p_t^i$ and using properties of RKHSs, we have for $t \in [0, 1]$:

$$\|\nabla L(v)_t\|^2 = \sum_{1 \leq i,j \leq N} (p_t^i)^\top K(z_t^i, z_t^j)p_t^j = \langle \tilde{p}_t, \mathbb{K}((z_t^i)_i))\tilde{p}_t \rangle,$$

where $\mathbb{K}$ is the kernel matrix associated with the points $(z_t^i)_i$. This last equality gives the result using Assumption 2 and the previous estimates on $p^i$. $\qquad\square$

Note that the degeneracy of the bounding functions $M, m$ as $R \to +\infty$ readily appears in Eq. (14). Thus one should not expect these bounds to imply global convergence of GD without making any further assumption. Indeed, cases where GD fails to converge towards a global optimizer of the loss are observed in [7], Section 6, with a setup corresponding to the model of Definition 1 with $V$ as in Eq. (3) and $\varphi = \mathrm{Id}_{\mathbb{R}^q}$. Also, note that the data separation $\delta$ plays an important role in Property 2 as it intervenes in the conditioning of the kernel matrix. In what follows, we always assume the data points to have a data separation lower-bounded by $\delta > 0$.

## 4.2 Convergence of RKHS-NODE

Thanks to the convergence analysis for overparameterized models detailed in Section 3, our main result follows as a consequence of the previous property. Theorem 2 is proven in Appendix C.3.

**Theorem 2.** *Let $V$ satisfy Assumption 1 with constant $\kappa$ and Assumption 2 with $\lambda, \Lambda$. Let $v^0$ be some initialization of the control parameter with $\|v^0\|_{L^2} = R_0$ and assume there exists a positive radius $R \geq 0$ s.t.:*

$$\frac{\sqrt{8}\sigma_{\max}(B^\top)\sqrt{N\Lambda L(v^0)}e^{3\kappa(R+R_0)}}{\sigma_{\min}(B^\top)^2\lambda(\sigma_{\min}(A)^{-1}\delta^{-1}e^{\kappa(R+R_0)})} \leq R. \tag{15}$$

*Then, for a sufficiently small step-size $\eta > 0$, GD with step-size $\eta$ converges towards a minimizer of the training loss at a linear rate and inside a ball of radius $R$. More precisely, for every $k \geq 0$:*

$$L(v^k) \leq (1 - \eta\mu)^k L(v^0), \quad and \quad \|v^k - v^0\|_{L^2} \leq R, \tag{16}$$

*where $\mu := \frac{1}{N}\sigma_{\min}^2(B^\top)\lambda\left(\sigma_{\min}^{-1}(A)\delta^{-1}e^{\kappa(R+R_0)}\right)e^{-2\kappa(R+R_0)}$.*

As Theorem 1, Theorem 2 is a local convergence result in which the condition in Eq. (15) expresses a threshold between two kinds of behaviours: **(i)** if $L(v^0)$ is sufficiently small, the training dynamic converges towards a global minimizer. The limiting behaviour is when the l.h.s. of Eq. (15) tends to 0. Because of a regularizing effect of GD (i.e. that $\|v^k - v^0\|_{L^2} \leq R$), the parameter stays in a ball of arbitrary small radius $R$ all along the training dynamic. In this limit, we recover a "linear" or "kernel" regime where the model is well approximated by its linearization at $v^0$ [14, 35, 26]. **(ii)** If $L(v^0)$ is too large, the result says nothing about the convergence of the GD. However, it is still observed in practice that the training dynamic often converges towards a global minimizer of the loss [60]. Explaining this phenomenon in a general setting remains a challenging open question.

## 5 Enforcing convergence with high dimensional embedding and finite width

As Theorem 2 is a local convergence result, it does not allow to conclude a general convergence behaviour of GD in the training of RKHS-NODE. In the following, we show how one can enforce the hypothesis of Theorem 2 to be verified and prove two global convergence results. The first one relies on suitably choosing matrices $A$ and $B$ in order to satisfy Eq. (15) and applies in the case of infinite width, i.e. with residual layers in a universal RKHS. The second result recovers global convergence in a finite width regime, relying on a high number $r$ of Random Fourier Features.

For the sake of readability we only consider here the case where $V$ belongs to a restricted class of RKHSs and refer to Appendix D for more general results and complete proofs. For some positive parameter $\nu > 0$ we consider the Matérn kernel $k$ defined in [57]:

$$\forall r \in \mathbb{R}_+, \; k(r) = \frac{2^{1-\nu}}{\Gamma(\nu)}\left(\frac{\sqrt{2\nu}}{2\pi}r\right)^\nu \mathcal{K}_\nu\left(\frac{\sqrt{2\nu}}{2\pi}r\right), \tag{17}$$

where $\Gamma$ is the Gamma function and $\mathcal{K}_\nu$ is the modified Bessel function of the second kind. Equivalently, $k$ can be defined by its frequency distribution over $\mathbb{R}^q$ as:

$$\forall x \in \mathbb{R}^q, \; k(\|x\|) = \int_{\mathbb{R}^q} e^{\imath\langle x,\omega\rangle}\mu_q(\omega)\mathrm{d}\omega \quad \text{with} \quad \mu_q(\omega) = C_{q,\nu}(1 + \frac{\|\omega\|^2}{2\nu})^{-(\frac{q}{2}+\nu)} \tag{18}$$

and $C_{q,\nu}$ a normalizing constant. For every $q \geq 1$, such a function is known to define a structure of vector-valued RKHS $V_q$ over $\mathbb{R}^q$ corresponding to the Sobolev space $H^{\nu+q/2}(\mathbb{R}^q, \mathbb{R}^q)$ [52, 57]. The associated kernel is given for every $z, z' \in \mathbb{R}^q$ by: $K_q(z, z') = k(\|z - z'\|)\,\mathrm{Id}_q$. Note that it is important for this RKHS to depend on the ambient dimension $q$. In particular the Sobolev space $H^s(\mathbb{R}^q, \mathbb{R}^q)$ is a RKHS if and only if it has regularity $s > q/2$. Assuming $\nu > 2$, $\mu_q$ further admits up to 4 finite order moment implying that $k$ is four times differentiable at 0 [28]. Then $V_q$ satisfies Assumption 1 with some constant $\kappa$ depending only on $\nu$ and given by Property 4:

$$\kappa = \sqrt{k(0)} + \sqrt{-k''(0)} + \sqrt{k^{(4)}(0)} = 1 + \sqrt{\frac{\nu}{\nu - 1}} + \sqrt{\frac{3\nu^2}{(\nu - 1)(\nu - 2)}}. \tag{19}$$

Also, $V_q$ satisfies Assumption 2 with $\Lambda \leq N$ and $\lambda$ depending a priori on $\nu$, $q$ and $N$.

Note that with this choice of scaling for $k$ and $\mu_q$, one recovers the Gaussian kernel $k : r \mapsto e^{-r^2/2}$ in the limit $\nu \to +\infty$ [57]. Thereafter we consider, $\nu \in (2, +\infty]$, the case $\nu = +\infty$ referring to the Gaussian kernel. We also assume that the data distribution is compactly supported. In particular there exists some $r_0 \geq 0$ so that every input data $x$ verifies $\|x\| \leq r_0$.

## 5.1 Global convergence with high-dimensional lifting

We first show how Eq. (15) can be satisfied by considering appropriate embedding matrices $A$ and $B$. Doing so, the square distance between the data points, i.e. the model's loss, is preserved whereas the conditioning of the kernel matrix can be controlled.

**Proposition 1.** *Let $\nu \in (2, +\infty]$, let $(x_i, y_i)_{1 \leq i \leq N} \in (\mathbb{R}^d \times \mathbb{R}^{d'})^N$ be a dataset with data separation $\delta > 0$ and let $R > 0$. There exist $q \geq 1$ and matrices $A \in \mathbb{R}^{q \times d}$, $B \in \mathbb{R}^{d' \times q}$ s.t. GD initialized at $v^0 = 0$ converges towards a zero-training-loss optimum in the training of RKHS-NODE. In particular, Eq. (15) holds with radius $R$ and $\kappa, \lambda, \Lambda$ associated with the RKHS $V_q$.*

As shown in the proof in Appendix D.1, Proposition 1 still holds for small but non-zero initialization. We present here two ways of obtaining matrices $A$ and $B$ satisfying Eq. (15):

**Scaling** Consider $A = \alpha(\mathrm{Id}_d, 0)^\top \in \mathbb{R}^{(d+d') \times d}$ and $B = \alpha(0, \mathrm{Id}_{d'}) \in \mathbb{R}^{d' \times (d+d')}$, for $\alpha > 0$. We show in Appendix D.1.2 that, in this setting, the l.h.s. of Eq. (15) scales as $\mathcal{O}(1/\alpha)$ and thus Theorem 2 holds for large enough $\alpha$. Moreover, observe that $q = d + d'$ is independent of $N$ and $\delta$ and such a regime can easily be implemented in practice. However, it has been shown that, although interpolation of the training data can be achieved as a consequence of a suitable rescaling of the parameters, this "lazy regime" can also lead to bad generalization properties [15].

**Lifting** Consider for $q \geq 1$ the matrices: $A_q := q^{-1/4}(\mathrm{Id}_d, ..., \mathrm{Id}_d, 0)^\top \in \mathbb{R}^{q \times d}$ and $B_q := q^{1/4}(\mathrm{Id}_{d'}, 0...0) \in \mathbb{R}^{d' \times q}$, with $\lfloor q/d \rfloor$ copies of $\mathrm{Id}_d$ in $A_q$. This choice is motivated by the intention for these matrices to produce a high-dimensional lifting, which has been shown to improve on the expressivity of ResNets [18]. We then show in Appendix D.1.1 that Eq. (15) can be satisfied for $q = \Omega(N^4 + \delta^{-4} \log(N)^4)$. We do not expect our condition on $q$ to be optimal as we observe in experiments (see Appendix A) that a regime of linear convergence can be obtained for $q \ll N^4 + \delta^{-4} \log(N)^4$. However, we observe that increasing $q$ does improve on the convergence and generalization properties of our model (Fig. 2).

## 5.2 Global convergence with finite width

In the preceding we showed that, in the case of an RKHS defined by a Matérn kernel, convergence of GD can be ensured for well-chosen matrices $A$ and $B$. However, for practical implementations, the form of the residual in Eq. (3) forces us to consider RKHSs defined by feature maps. A way to overcome this difficulty and to benefit from the properties of a wide range of kernel functions is to consider an approximation by *Random Fourier Features (RFF)* [48, 49]. More precisely, given $q \geq 1$, recall the definition of the Matérn kernel $k$ in terms of its frequency distribution $\mu_q$ over $\mathbb{R}^q$ in Eq. (18) and for any sampling $\omega^1, ..., \omega^r \overset{iid}{\sim} \mu_q$ of size $r$, consider the feature map:

$$\varphi : z \in \mathbb{R}^q \mapsto \frac{1}{\sqrt{r}}(e^{\imath \langle z, \omega^j \rangle})_{1 \leq j \leq r} \in \mathbb{C}^r. \tag{20}$$

In other words, considering the complex activation $\sigma : t \mapsto e^{\imath t}$ applied component-wise and $U := (\omega^1 | \ldots | \omega^r) \in \mathbb{R}^{q \times r}$ the feature matrix, we have $\varphi(z) = r^{-1/2}\sigma(U^\top z)$. Recall that such a feature map defines a structure of RKHS on $\hat{V}_q := \{W\varphi(.) \mid W \in \mathbb{R}^{q \times r}\}$. Such a $\hat{V}_q$ can be viewed as a finite-dimensional approximation of the universal RKHS $V_q$ as it is associated with the kernel function $\hat{K}_q(z, z') := \hat{k}(z, z') \, \mathrm{Id}_q$, with:

$$\hat{k}(z, z') := \langle \varphi(z), \varphi(z') \rangle = \frac{1}{r} \sum_{j=1}^{r} e^{\imath \langle z - z', \omega^j \rangle} \xrightarrow{r \to +\infty} k(\|z - z'\|) \text{ a.s.}$$

Given any $q \geq 1$, we show that, with high probability over the choice of features, $\hat{V}_q$ recovers the properties of admissibility and universality of $V_q$ as soon as $r$ is sufficiently high w.r.t. $q$ and $N$. The following is a particular case of Proposition 5 in Appendix D.2.

**Proposition 2.** *Consider any $q, N \geq 2$ and any $\epsilon, \tau, R > 0$. Assume $\nu > 4$.*
*(i) For $r \geq \Omega(\tau q^8)$, with probability greater than $1 - \tau^{-1}$, $\hat{V}_q$ satisfies Assumption 1 with $\hat{\kappa} \leq \kappa + 1$.*
*(ii) For $r \geq \Omega(\epsilon^{-2} N^2 (q \log(\|A\|_2 r_0 + R) + \tau))$, with probability greater than $1 - e^{-\tau}$, for any $v \in L^2(\hat{V}_q)$ s.t. $\|v\|_{L^2} \leq R$ and any time $t \in [0,1]$: $\lambda_{\min}(\hat{\mathbb{K}}((z_t^i)_i)) \geq \lambda_{\min}(\mathbb{K}((z_t^i)_i)) - \epsilon$, where $(z^i)_i$ are the solutions to Eq. (6) and $\hat{\mathbb{K}}$, $\mathbb{K}$ are the kernel matrices of $\hat{k}$ and $k$ respectively.*

*Sketch of proof for (i).* First note that for $\nu > 4$, $\mu_q$ admits up to $8^{th}$-order finite moments and these can be bounded uniformly in $q$ [28]. Let $\varphi$ be the feature map of Eq. (20). Then for every $z \in \mathbb{R}^q$, $\|\varphi(z)\| \leq 1$ so that for every $v \in \hat{V}_q$, $\|v\|_\infty \leq \|W\|\|\varphi\|_\infty \leq \|v\|_V$. For the differential $Dv$ we have for every $z \in \mathbb{R}^q$:

$$D\varphi(z) = \frac{1}{\sqrt{r}} \left( \omega_i^j e^{-\iota \langle z, \omega^j \rangle} \right)_{i,j} \in \mathbb{R}^{r \times q}.$$

Then, by the Bienayme-Chebyshev inequality, $D\varphi(z)^* D\varphi(z) = \frac{1}{r} \sum_{j=1}^r \omega^j (\omega^j)^\top$ converges in probability to $-k''(0) \operatorname{Id}_q$ as $r \to +\infty$. Thus, for $\alpha > 0$ and $r$ sufficiently high w.r.t. $q$, $\alpha$ and $\tau$, $\|Dv\|_{2,\infty} = \|WD\varphi\|_{2,\infty} \leq \sqrt{-k''(0) + \alpha} \|v\|_{\hat{V}_q}$, with probability greater than $1 - \tau^{-1}$. The same idea applies to bound $\|D^2 v\|_{2,\infty}$ and the result follows using that $\kappa$ is given by Eq. (19).

$\square$

Finally, combining Proposition 1 and Proposition 2, we obtain a global convergence result. Theorem 3 states convergence, with high probability over a choice of features, of GD towards a zero-training-loss optimum for infinitely deep ResNets of finite width.

**Theorem 3** (Global convergence). *Assume $\nu > 4$ and let $(x^i, y^i) \in (\mathbb{R}^d \times \mathbb{R}^{d'})^N$ be a compactly supported dataset with input data separation $\delta > 0$. There exist matrices $A \in \mathbb{R}^{q \times d}$ and $B \in \mathbb{R}^{d' \times q}$ s.t. for any $\tau > 0$, with probability at least $1 - \tau^{-1}$ w.r.t. the choices of features, GD initialized at $v^0 = 0$ converges towards a zero training loss optimum in the training of the RKHS-NODE model of Definition 1 with the feature map $\varphi$ of Eq. (20) as soon as $r \geq \Omega(\tau(q^8 + qN^2 \log(\|A\|_2))$.*

*Proof.* Consider $R = 1$. By Proposition 1, we can have $A \in \mathbb{R}^{q \times d}$, $B \in \mathbb{R}^{d' \times q}$ so that in Eq. (15):

$$\frac{8\sqrt{2}\sigma_{\max}(B^\top)\sqrt{N\Lambda L(0)}e^{3(\kappa+1)}}{\sigma_{\min}(B^\top)^2 \lambda(\sigma_{\min}(A)^{-1}\delta^{-1}e^{(\kappa+1)})} \leq 1,$$

for $\kappa$, $\lambda$ and $\Lambda$ associated with $k$. Also, by the proof of Proposition 1 we can have: $\lambda(\sigma_{\min}(A)^{-1}\delta^{-1}e^{(\kappa+1)}) \geq 1/2$. Taking $\epsilon = 1/4$ in Proposition 2, the condition in Eq. (15) is satisfied by $\hat{V}_q$ with probability greater than $1 - \tau^{-1}$ as soon as $r \geq \Omega(\tau q^8 + \tau qN^2 \log(1 + \|A\|_2 r_0))$. $\square$

## 6 Conclusion

We have identified a relevant infinite width limit (RKHS-NODE) for a particular model of ResNet. We showed that GD converges linearly when training this model and that a network's width polynomial w.r.t. to the size of the dataset is sufficient to maintain this property. A natural extension of our result is to study the convergence of GD when also training the hidden layers of the residuals. A first step towards this general case consists in studying the corresponding mean field model where the residuals are parameterized by density distributions over the neurons [14, 41, 40, 27, 38, 21] for each residual blocks. Interestingly, such a parametrization of the residual blocks is still linear in this measure and thus fits into our framework of linear in parameters. However, it would require a finer mathematical analysis to obtain similar results.

**Potential Negative Societal Impacts.** Our work aims at improving the theoretical and practical understanding of deep networks and therefore we do not expect a direct negative impact.

## Acknowledgements

The work of Gabriel Peyré was supported by the French government under management of Agence Nationale de la Recherche as part of the "Investissements d'avenir" program, reference ANR19-P3IA-0001 (PRAIRIE 3IA Institute) and by the European Research Council (ERC project NORIA).

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
