# A Numerical experiments

The goal of this section is to quantify how much (in addition to interpolating the training dataset) our model is able to generalize on the test dataset. This is also useful to compare the performances of our model with those of standard ResNet architectures (which integrate batch normalization and training of the hidden layers). We implemented our model in Pytorch [45] and trained it on image datasets for classification tasks. Source code is available at `https://github.com/rbarboni/FlowResNets`.

Experiments were conducted using a private infrastructure, which has a carbon efficiency of 0.05 kgCO$_2$eq/kWh. A cumulative of (at most) 1000 hours of computation was performed on hardware of type Tesla V100-PCIE-16GB (TDP of 300W). Total emissions are estimated to be 15 kgCO$_2$eq (or 60km in an average car) of which 0 percents were directly offset.
Estimations were conducted using the MachineLearning Impact calculator presented in [30].

**Computational setup for classification tasks.** In the context of classification tasks, we use a cross entropy loss in place of the least square loss of Eq. (7). For a problem with $K$ classes, the output dimension of the model is $d' = K$ and targets $y \in \mathbb{R}^K$ are one-hot vector encoding the target classes. For a batch of $N$ predictions $(z^i)_{1 \leq i \leq N}$ and targets $(y^i)_{1 \leq i \leq N}$ in $\mathbb{R}^K$ the *Cross Entropy* loss is defined as:

$$\texttt{CrossEntropy}((z^i)_i, (y^i)_i) \coloneqq \frac{1}{N} \sum_{i=1}^{N} \ell(z^i, y^i),$$

where $\ell$ is the *Binary Entropy* defined for one prediction $z$ and one target $y \in \mathbb{R}^K$ by:

$$\ell(z, y) \coloneqq \frac{\sum_{j=1}^{K} y_j^i e^{z_j^i}}{\sum_{j=1}^{K} e^{z_j^i}}.$$

Then for a model $F$ depending on the parameters $W$ and a training batch $(x^i, y^i)_{1 \leq i \leq N}$ we define the empirical risk:

$$L(W) \coloneqq \texttt{CrossEntropy}((F(W, x^i)_i, (y^i)_i),$$

and train the model by *Stochastic Gradient Descent (SGD)* on $W$. Finally, the performance of the model is assessed by the *Top-1 error rate* on a test dataset.

Note that, as explained in Remark 2, the result of Theorem 3 can be extended to this cross entropy loss. Indeed, $\ell$ satisfies a functional inequality similar to the Polyak-Lojasiewicz inequality. Assuming without loss of generality that $y = e_1$ is the indicator of class 1, one has:

$$\nabla_{z_1} \ell(z, y) = e^{-\ell(z,y)} - 1,$$

Then by convexity of exponential, when $\ell(z, y) \leq 1$:

$$\|\nabla_z \ell(z, y)\|^2 \geq (1 - e^{-\ell(z,y)})^2 \geq (1 - e^{-1})^2 \ell(z, y)^2.$$

Note however that Theorem 3 is only valid for full batch gradient descent. We leave its extension to SGD for future works.

## A.1 Experiments on MNIST

We implemented the model of Definition 1 on Pytorch using the `torchdiffeq` package [11] and performed experiments on the MNIST dataset.

**Implementation using `torchdiffeq`.** The model of Definition 1 is implemented as a succession of convolutional layers. Given some number of layers $L$ the trained parameters consist of convolution matrices $W_k \in \mathbb{R}^{C \times C_{int} \times 3 \times 3}$ for $k \in [\![0, L]\!]$, with $C$ the number of channels of the input image and $C_{int}$ some number of channels for the hidden layers. The control parameter $v$ is defined at discrete time steps $\{k/L\}_{0 \leq k \leq L}$ by:

$$v_{k/L}(x) = W_k \star \texttt{ReLU}(U \star x),$$

where $U \in \mathbb{R}^{C_{int} \times C \times 3 \times 3}$ is a fixed and untrained convolution matrix. We refer to this setting as a ResNet with RKHS residuals. On the other hand, we refer to the setting where $U$ is replaced at each layer by trained convolution matrices $U_k$ as ResNet with *Single Hidden Layer (SHL)* residuals.

**Remark 6.** *By analogy with the definition of RKHSs generated by random features (Eq. (20)), the ratio between the number of features and the dimension is here:*

$$\frac{r}{q} = \frac{C_{int}}{C}.$$

For any $t \in [0, 1]$, $v_t$ is defined by affine interpolation:

$$v_t(x) := v_{k/L}(x) + (tL - k)\left(v_{(k+1)/L}(x) - v_{k/L}(x)\right)$$
$$= (W_k + (tL - k)(W_{k+1} - W_k)) \star \sigma(U \star x),$$

with $k = \lfloor tL \rfloor$. The forward method consists in integrating the ODE of Eq. (6) with control parameter $v$ using the `torchdiffeq.odeint` method [11]. For some input $z_0$ define:

$$z_1((W_k), z_0) := \texttt{torchdiffeq.odeint}(v, z_0, [0, 1]),$$

then for an image input $x$ the model's output is given by:

$$F((W_k), x) = \texttt{B}(z_1((W_k), \texttt{A}(x))),$$

where `A` and `B` are small convolutional networks, fixed during the training of $F$. These networks play the same role as the matrices $A$ and $B$ in Definition 1, that is they are used for the purpose of adjusting the data dimension.

**Hyperparameter tuning.** Several choices of hyperparameters can affect the performances of the model.

- The convolution matrix $U$: as detailed in Section 5, the way the weights of $U$ are sampled determines to which RKHS $V$ belongs the control parameter $v$. For the sake of simplicity we choose to sample the coefficients of $U$ as i.i.d. Gaussians.

- The initialization of $(W_k)$: the weights of the convolution matrices $W_k$ are initialized to $0$. For an input image $x$ the output is given at initialization by $F(0, x) = \texttt{B}(\texttt{A}(x))$.

- The integration method: `torchdiffeq.odeint` allows the user to choose an integration method. We observed an *explicit midpoint* method to offer a good trade-off between performance and numerical stability w.r.t. other fixed-steps methods such as *explicit Euler* or *RK4*.

- The number of layers $L$: we tested our model for $L \in \{5, 10, 20\}$. This parameter controls the total number of parameters of the model.

- The networks `A` and `B`: their choice defines the dimension of space in which the forward ODE Eq. (6) is integrated, which is expected to have an important impact on the performances of the model (c.f. Section 5). Moreover, as the parameters $(W_k)$ are initialized at $0$, the performances of the model before training are those of the concatenation $\texttt{B} \circ \texttt{A}$. Without training, the classification error of $\texttt{B} \circ \texttt{A}$ is of $90\%$ while with enough training, it can be as good as $2\%$. We tested our model with different levels of training of $\texttt{B} \circ \texttt{A}$.

**Results.** Figure 1 shows the evolution of the performances of RKHS-NODEs while trained on the MNIST dataset. The decay of the Empirical Risk is directly related to the decay of the classification error. Without pretraining `A` and `B`, our model already achieves up to $98\%$ accuracy on the test set. When `A` and `B` are pretrained RKHS-NODE still improves on the starting accuracy: in this setting more than $99\%$ accuracy is reached. Most importantly, Fig. 1 shows that not training the hidden layers inside residual blocks does not significantly hinders the performances in classification. Indeed, comparing the performances of ResNets with RKHS residuals and SHL residuals one observes a $1\%$ accuracy drop when training RKHS-NODE from scratch (Fig. 1a) and $0.5\%$ accuracy when networks `A` and `B` are pretrained (Fig. 1b).

Finally we showcase the relevance of the analysis of Section 5 by training our model with a varying number of input channels in Fig. 2. We observe a significant drop in convergence of the empirical risk with $4$ channels compared with $8$ and $32$ channels. Non-convergence of the empirical risk also implies poorer performances in generalization. Such results are coherent with the convergence condition of Eq. (15): augmenting the data dimension allows to have global convergence when the loss at initialization is too high.

## A.2 Experiments on CIFAR10

We performed experiments on the CIFAR10 dataset, using an architecture inspired from ResNet18 [25].

**Implementation.** Our architecture relies on the ResNet18 architecture [25] but residual blocks are changed and simplified (by removing the final non-linearity and the batch-normalization) to match the definition of RKHS-NODE (Definition 1). Each residual block consists in the composition of a convolution $U$, a ReLU non-linearity and a convolution $W$. More precisely, for an input image $x$, the output of the $k^{th}$ layer reads:

$$\mathcal{F}_k(x) = x + W_k \star \texttt{ReLU}(U_k \star x),$$

where $U_k \in \mathbb{R}^{C_{int} \times C \times 3 \times 3}$, $W_k \in \mathbb{R}^{C \times C_{int} \times 3 \times 3}$ are convolution matrices, $C$ is the number of channels of the input image and $C_{int}$ is the number of channels of the hidden layer. When both convolutions $W_k$ and $U_k$ are trained, we refer to these residuals as *Single Hidden Layer (SHL) residuals*. In the framework of RKHS-NODE, all convolutions $U_k$ are fixed and set to the same convolution $U$. We refer to it as *RKHS residuals*.

Finally, ResNet18 consists of 4 blocks each containing 2 residual layers. We keep 2 of our residuals in the first, second and fourth block but stack an arbitrary number $D$ of residual layers in the third block. Thereby we refer to this third block as the NODE block, which performs the integration of Eq. (6).

Note that compared to the residuals in the original ResNet18 architecture, batch-normalization at input and output of the residuals as well as ReLU non-linearities are removed. Moreover, in order to reproduce the framework of *Random Fourier Features* (Eq. (20)), the weights of $U$ are sampled as i.i.d. gaussians and rescaled by a $C_{int}^{-1/2}$ factor. Finally, the weights of the convolutions $W_k$ are initialized at $0$. Such an initialization corresponds in many ways to the one proposed in [61].

**Results.** Fig. 3 reports the training of RKHS-NODE on the CIFAR10 dataset. Figure 3a shows the training of RKHS-NODE (RKHS residuals) and is to be compared with Fig. 3b which shows the training of the same model but with trained hidden layers (SHL residuals). Our experiments show that similar performances can be achieved: both ResNets achieve up to $88\%$ accuracy on the test dataset. As a comparison, the ResNet18 original architecture can be trained to achieve up to $94\%$ accuracy.

Finally, Fig. 3 also compares the performances of the model depending on the number of layers inside the NODE block. One observes significantly different behavior when there is no NODE (1 layer) and one there is (10 and 20 layers): more layers are related to better performances both on the train dataset and on the test dataset and both when hidden layers are trained or not. However, one sees that the improvement related to adding more layers is limited: performances with 10 and 20 layers are very similar and a NODE block with 1 layers already achieves $82\%$ accuracy RKHS residuals and $84\%$ accuracy with SHL residuals.

# B  Proofs of Section 3

We give a proof of Theorem 1. This essentially follows the proof given in [36].

*Proof of Theorem 1.* Assume the loss $L$ satisfies Definition 2 with $M$ and $m$ and that Eq. (9) is satisfied at initialization $v^0 \in \mathbb{R}^m$. The proof proceeds by induction over the gradient step $k$

Assume the convergence rate and the regularization bound of Eq. (10) are satisfied for every $l \leq k$. Then at step $k + 1$:

$$\|v^{k+1} - v^0\| = \|\eta \sum_{l=0}^{k} \nabla L(v^l)\| \leq \eta \sum_{l=0}^{k} \|\nabla L(v^l)\|$$

$$\leq \eta \sum_{l=0}^{k} \sqrt{2M(\|v^l\|)L(v^l)}.$$

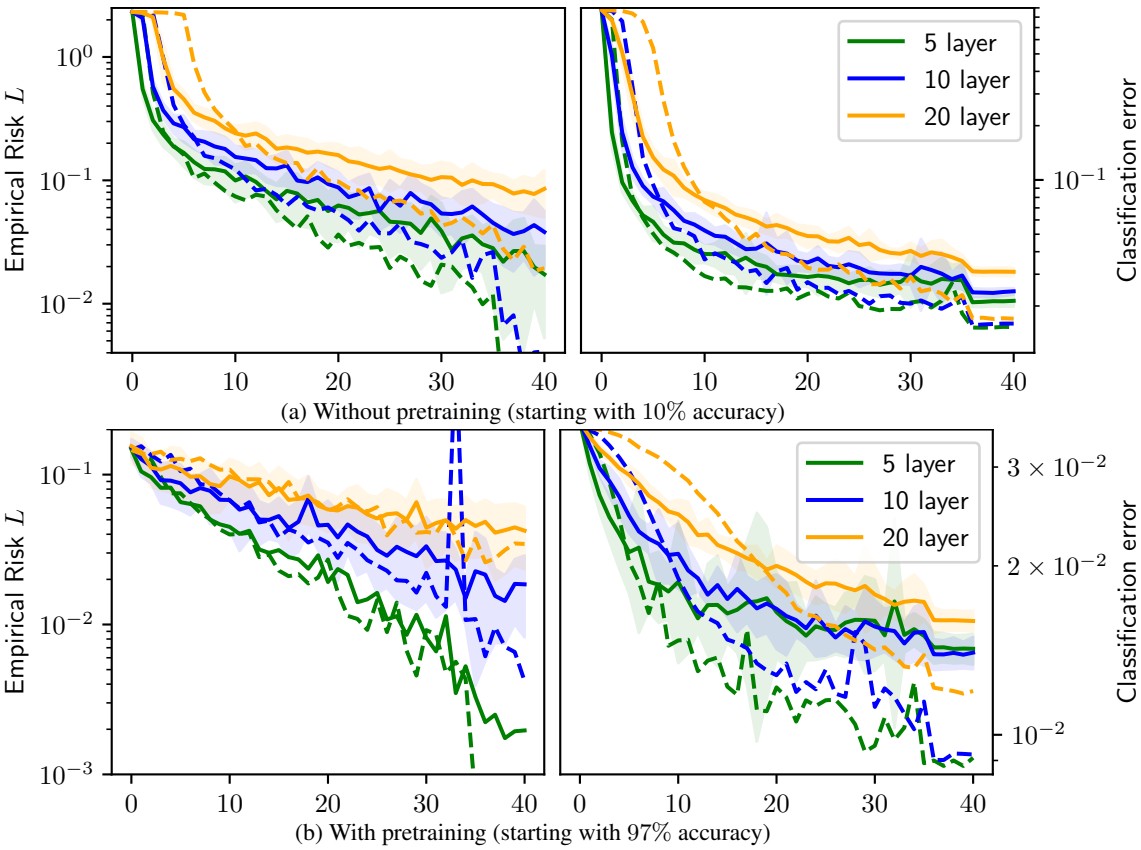

Figure 1: Performances of NODE with 32 channels while trained on MNIST with SGD. Left column reports evolution of the empirical risk and right column reports evolution of classification error, both for ResNets with RKHS residuals (plain) and SHL residuals (dashed). The $x$-axis is the number of pass through the dataset. Experiments are performed with different levels of pretraining of A and B, corresponding to different starting accuracy ((a)-(b)), and with different number of layers. Learning rate and batch size are fixed, learning rate is divided by 10 after 35 iterations. Plots are average over 20 runs, lines are means and, for RKHS residuals, colored areas are mean $\pm$ one standard deviation.

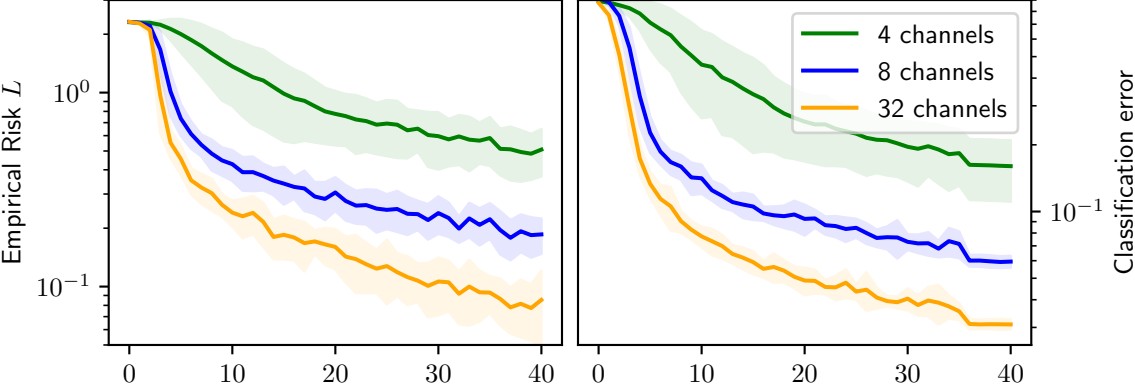

Figure 2: Training of RKHS-NODE on MNIST with 20 layers, 4, 8 and 32 input channels $C$ and without pretraining. The $x$-axis is the number of pass through the dataset. The rate $C_{int}/C = 1$ is the same for each model. Learning rate and batch size are fixed, learning rate is divided by 10 after 35 iterations. Plots are average over 20 runs, lines are means and colored areas are mean $\pm$ one standard deviation.

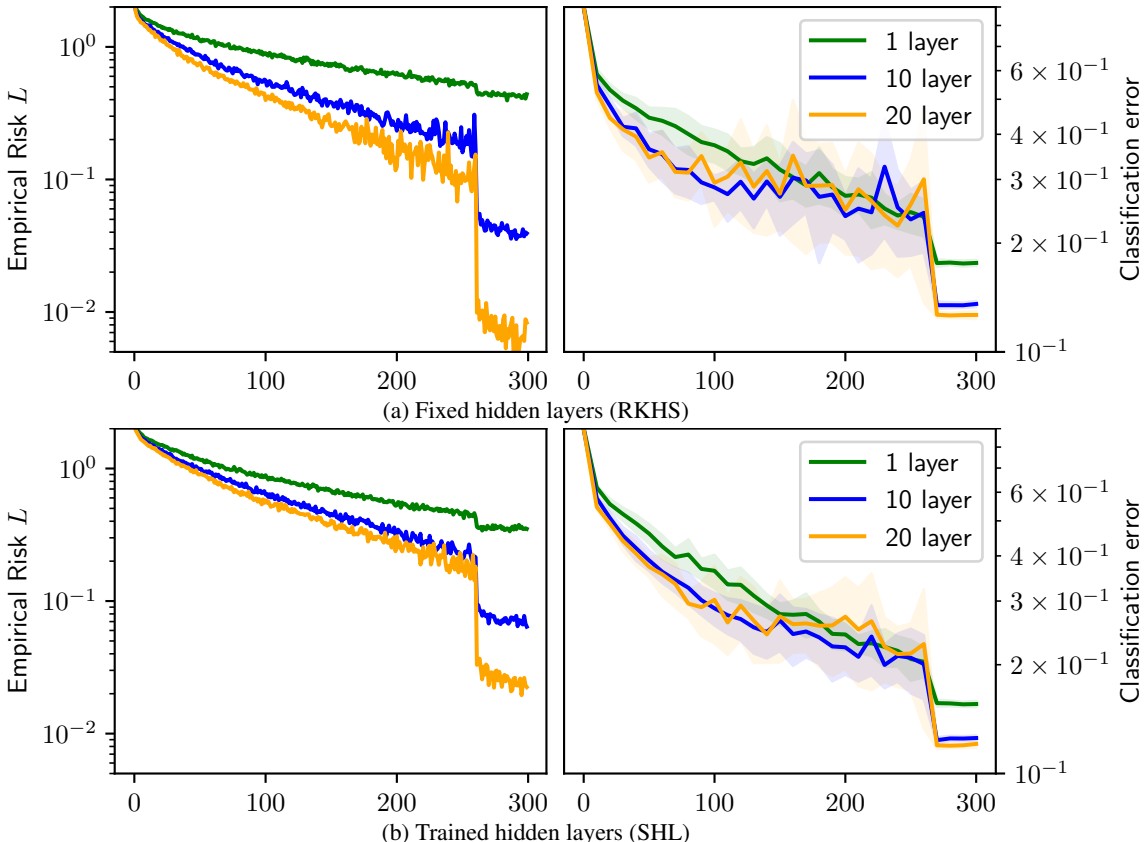

Figure 3: Performances of RKHS-NODE while trained on CIFAR10 with SGD (256 images per batch). Left column reports evolution of the empirical risk on the train set and right column reports the classification error on the test set. The $x$-axis is the number of pass through the dataset. Learning rate and batch size are fixed, learning rate is divided by 10 after 260 iterations. Plots are average over 20 runs, lines are means and colored areas are mean $\pm$ one standard deviation.

Using the induction hypothesis and setting $\mu = m(\|v^0\| + R)$ we have:

$$
\begin{aligned}
\|v^{k+1} - v^0\| &\leq \eta \sqrt{2M(\|v^0\| + R)L(v^0)} \sum_{l=0}^{k} (1 - \eta\mu)^{-l/2} \\
&\leq \eta \sqrt{2M(\|v^0\| + R)L(v^0)}(1 - \sqrt{1 - \eta\mu})^{-1} \\
&\leq \frac{2}{\mu} \sqrt{2M(\|v^0\| + R)L(v^0)} \\
&\leq R,
\end{aligned}
$$

where the last inequality is Eq. (9). We thus recovered the regularization bound of Eq. (10) at step $k + 1$.

Moreover, because $v^{k+1}$ is located in $B(v^0, R)$ we have thanks to the smoothness assumption:

$$
\begin{aligned}
L(v^{k+1}) &\leq L(v^k) - \eta\|\nabla L(v^k)\|^2 + \eta^2 \frac{\beta}{2}\|\nabla L(v)\|^2 \\
&\leq L(v^k) - \frac{\eta}{2}\|\nabla L(v^k)\|^2,
\end{aligned}
$$

because $\eta \leq \beta^{-1}$. Thus using the lower bound in the PL inequality Eq. (8):

$$
L(v^{k+1}) \leq L(v^k)(1 - m(\|v^0\| + R)\eta),
$$

which gives the convergence rate of Eq. (10) at step $k + 1$ by induction on $k$.

$\square$

# C  Proofs of Section 4

## C.1  About the definition of RKHS-NODE

Before deriving proofs for the properties of our RKHS-NODE model, it is interesting to study carefully the well-posedness of Definition 1. Indeed, because the control parameter $v$ is only integrable in time and not continuous, the Cauchy-Lipschitz theorem does not ensure that there exist solutions to Eq. (6). Instead we rely on a weaker notion of solution and use a result from Carathéodory (Section I.5 in [23]).

**Proposition 3.** *Let V be some RKHS satisfying Assumption 1 and $v \in L^2([0,1], V)$ be some control parameter. Then for every $x \in \mathbb{R}^d$ there exists a unique solution $z$ of Eq. (6) in the weak sense of absolutely continuous functions. More precisely there exists a unique $z \in H^1([0,1], \mathbb{R}^q)$ such that for every $t \in [0,1]$:*

$$z_t = Ax + \int_0^t v_s(z_s)ds. \tag{21}$$

*Proof.* The map $(t, z) \in [0,1] \times \mathbb{R}^q \mapsto v_t(z)$ is measurable and by Assumption 1 we have for every $t \in [0,1]$ and every $z \in \mathbb{R}^q$:

$$\|v_t(z)\| \le \kappa \|v_t\|_V,$$

whose upper-bound is integrable w.r.t. $t \in [0,1]$. Then, applying Theorem 5.1 of [23] gives a unique absolutely continuous solution $z$ of Eq. (21). Applying Assumption 1 once again, we have that $\dot{z}$ is square integrable and thus $z$ is in $H^1$. $\square$

In the paper, every equality implying derivatives has to be understood in the sense of weak derivatives of $H^1$ functions. In particular, this notion allows to perform integration by parts, which is used in the following proof of Property 1.

*Proof Property 1.* Consider the optimization problem of minimizing the empirical risk of Eq. (7) with $F$ the RKHS-NODE model of Definition 1 and a dataset $(x^i, y^i)_{1 \le i \le N} \in (\mathbb{R}^d \times \mathbb{R}^{d'})^N$. Introducing for every index $i \in [\![1, N]\!]$ the variables $z^i \in H^1([0,1], \mathbb{R}^q)$ solutions of Eq. (6), this can be viewed as an optimisation problem over $((z^i)_i, v)$ under the constraint that Eq. (6) is satisfied:

$$\min_{\substack{(z^i)_i \in H^1(\mathbb{R}^q)^N \\ v \in L^2(V)}} \frac{1}{2N} \sum_{i=1}^N \|Bz_1^i - y^i\|^2$$

$$\text{with } \forall i \in [\![1, N]\!], \left\{ \begin{array}{rcl} \dot{z}_t^i & = & v_t(z_t^i) \ \forall t \in [0,1] \\ z_0^i & = & Ax^i. \end{array} \right.$$

Introducing the adjoint variables $(p^i)_i \in H^1(\mathbb{R}^q)^N$, the Lagrangian of the optimization problem is defined as:

$$\mathcal{L}((z^i), (P^i), v) := \sum_{i=1}^N \left( \frac{1}{2N} \|Bz_1^i - y^i\| + \int_0^1 \langle p_t^i, \dot{z}_t^i - v_t(z_t^i) \rangle \mathrm{d}t \right)$$

$$= \sum_{i=1}^N \left( \frac{1}{2N} \|Bz_1^i - y^i\| + \left[ \langle p_t^i, z_t^i \rangle \right]_0^1 - \int_0^1 \langle \dot{p}_t^i, z_t^i \rangle \mathrm{d}t - \int_0^t \langle p_t^i, v_t(z_t^i) \rangle \mathrm{d}t \right),$$

where the second equality is established by integration by parts. Therefore, the condition for optimality over $z^i$ is equivalent to Eq. (11). For every index $i$:

$$\nabla_{z^i} \mathcal{L} = 0 \Leftrightarrow \left\{ \begin{array}{rcl} \dot{p}_t^i & = & -Dv_t(z_t^i)p_t^i \\ p_1^i & = & -\frac{1}{N} B^\top (Bz_1^i - y^i), \end{array} \right.$$

which has to be understand in the sense of weak solutions in $H^1$.

The gradient of $L$ is obtained by differentiating over the $v$ variable. Denoting $\delta_z^p$ the linear form $v \mapsto \langle v(z), p \rangle$, we have:

$$\nabla L(v) = \nabla_v \mathcal{L}((z^i), (p^i), v)$$

$$= -\sum_{i=1}^{N} K * \delta_{z^i}^{p^i}$$

$$= -\sum_{i=1}^{N} K(., z^i) p^i,$$

with $K$ the kernel function of the RKHS $V$ and $K* : V^* \to V$ the associated isometry[1].

$\square$

## C.2  Proof of Property 2

We prove here that for any given dataset $(x^i, y^i)_{1 \le i \le N} \in (\mathbb{R}^d \times \mathbb{R}^{d'})^N$, the empirical risk $L$ associated with the RKHS-NODE model satisfies a (local) Polyak-Lojasiewicz property. As stated in Property 2. The proof uses Assumption 1 to derive estimates on the solutions of Eq. (6) and Eq. (11), which we give in the following lemma:

**Lemma 1.** *Let $V$ satisfy Assumption 1 with constant $\kappa$ and let $v \in L^2([0,1], V)$ be some control parameter.*

*(i) Let $(z^i)_{1 \le i \le N}$ be the solutions of Eq. (6) for some data inputs $(x^i)_{1 \le i \le N} \in (\mathbb{R}^d)^N$. Then for every indices $i, j \in [\![1, N]\!]$ and every time $t \in [0, 1]$:*

$$\|z^i - z^j\| \ge \sigma_{\min}(A) e^{-\kappa \|v\|_{L^2}} \|x^i - x^j\|. \tag{22}$$

*(ii) Let $(p^i)_{1 \le i \le N}$ be the solutions of Eq. (11) associated with $(z^i)_{1 \le i \le N}$ with objective outputs $(y^i)_{1 \le i \le N} \in (\mathbb{R}^{d'})^N$. Then for every $i \in [\![1, N]\!]$ and every time $t \in [0, 1]$:*

$$\frac{\sigma_{\min}(B^\top)}{N} e^{-\kappa \|v\|_L^2} \|B z_1^i - y^i\| \le \|p_t^i\| \le \frac{\sigma_{\max}(B^\top)}{N} e^{\kappa \|v\|_L^2} \|B z_1^i - y^i\|.$$

*Proof of Lemma 1.* **Proof of (i)** Let $i, j \in [\![1, N]\!]$. Assume by contradiction that for some time $t \in [0, 1]$ we have:

$$\|z_t^i - z_t^j\| < e^{-\kappa \|v\|_{L^2}} \|z_0^i - z_0^j\|.$$

Then because $z^i$ and $z^j$ are absolutely continuous, $\|z^i - z^j\|^2$ is absolutely continuous and for any time $s \in [0, 1]$:

$$\|z_s^i - z_s^j\|^2 = \|z_t^i - z_t^j\|^2 + 2 \int_t^s \langle v_r(z_r^i) - v_r(z_r^j), z_r^i - z_r^j \rangle \mathrm{d}r$$

$$\le \|z_t^i - z_t^j\|^2 + 2 \int_t^s \kappa \|v_r\|_V \|z_r^i - z_r^j\|^2 \mathrm{d}r,$$

where the inequality follows from $\|Dv_r\|_{2,\infty} \le \kappa \|v_r\|_V$. Applying Grönwall's lemma, we have:

$$\|z_s^i - z_s^j\|^2 \le \|z_t^i - z_t^j\|^2 e^{2\kappa \|v\|_{L^2}},$$

and by setting $s = 0$:

$$\|z_0^i - z_0^j\|^2 \le \|z_t^i - z_t^j\|^2 e^{2\kappa \|v\|_{L^2}} < \|z_0^i - z_0^j\|,$$

which is a contradiction. Therefore for any time $t \in [0, 1]$:

$$\|z_t^i - z_t^j\| \ge e^{-\kappa \|v\|_{L^2}} \|z_0^i - z_0^j\|,$$

and the result follows by considering the initial condition $z_0^i = Ax^i$.

---

[1] The notation $K*$ reminds of convolution which is the case when the kernel is translation invariant.

**Proof of (ii)** Let $i \in [\![1, N]\!]$ be any index and let $p^i$ be the solution of Eq. (11) with initial condition $p_1^i = -\frac{1}{N} B^\top (B z_1^i - y^i)$. Then because $p^i$ is absolutely continuous, $\|p^i\|$ is absolutely continuous and for any time $t \leq s \in [0, 1]$:

$$\|p_t^i\|^2 = \|p_1^i\|^2 - 2 \int_1^t \langle D v_s(z_s^i) p_s^i, p_s^i \rangle \mathrm{d}s,$$

so that using Assumption 1 we have:

$$\|p_s^i\|^2 \leq \|p_t^i\|^2 + 2 \int_t^s \kappa \|v_r\|_V \|p_r^i\|^2 \mathrm{d}r \,.$$

Using Grönwall's lemma in the first inequality and setting $s = 0$ we have:

$$\|p_1^i\|^2 \leq \|p_t^i\|^2 e^{2\kappa \|v\|_{L^2}},$$

and proceeding by contradiction (such as in (i)) we have:

$$\|p_1^i\|^2 \geq \|p_t^i\|^2 e^{-2\kappa \|v\|_{L^2}}.$$

The result follows by considering the initial condition on $p_1^i$. $\qquad\qquad\square$

Provided those estimates on $z^i$ and $p^i$, it remains to use Assumption 2 in order to conclude.

*Proof of Property 2.* Let $v \in L^2([0, 1], V)$ and consider the form of the gradient of $L$ given by Property 1 with $(z^i)_{1 \leq i \leq N}$ the solutions of Eq. (6) and $(p^i)_{1 \leq i \leq N}$ the solutions of Eq. (11). Let $t \in [0, 1]$, then by definition of the norm in RKHSs:

$$\|\nabla L(v)_t\|_V^2 = \sum_{1 \leq i, j \leq N} (p_t^i)^\top K(z_t^i, z_t^j) p_t^j \,,$$

where we recall that $K$ is the kernel associated with $V$. Noting $p := (p_t^i) \in \mathbb{R}^{Nq}$, the vector of the stacked $(p_t^i)_{1 \leq i \leq N}$, and $\mathbb{K}$ the kernel matrix associated with the family of points $(z_t^i)_i$, we have:

$$\|\nabla L(v)_t\|_V^2 = \langle p, \mathbb{K}p \rangle \,.$$

Then by Assumption 2, there exists a non-increasing function $\lambda$ and a constant $\Lambda$ such that:

$$\lambda(\max_{1 \leq i, j \leq N} \|z_t^i - z_t^j\|^{-1}) \|p\|^2 \leq \|\nabla L(v)_t\|_V^2 \leq \Lambda \|p\|^2.$$

Using (i) in Lemma 1 we have:

$$\lambda(\max_{1 \leq i, j \leq N} \|z_t^i - z_t^j\|^{-1}) \geq \lambda(\sigma_{\min}(A)^{-1} \delta^{-1} e^{\kappa \|v\|_{L^2}}),$$

where $\delta := \min_{1 \leq i, j \leq N} \|x^i - x^j\|$ is the data separation. Finally the result follows by using (ii). More precisely:

$$\|p\|^2 = \sum_{i=1}^N \|p_t^i\|^2$$

$$\leq \frac{\sigma_{\max}(B^\top)^2}{N^2} e^{2\kappa \|v\|_{L^2}} \sum_{i=1}^N \|B z_1^i - y^i\|^2$$

$$= 2 \frac{\sigma_{\max}(B^\top)^2}{N} e^{2\kappa \|v\|_{L^2}} L(v),$$

and in the same manner:

$$\|p\|^2 \geq 2 \frac{\sigma_{\min}(B^\top)^2}{N} e^{-2\kappa \|v\|_{L^2}} L(v).$$

$\qquad\qquad\square$

## C.3 Proof of Theorem 2

Theorem 2 is a direct consequence of Property 2. In order to apply Theorem 1, it suffices to show that $L$ satisfies some smoothness assumption as defined in Definition 3:

**Property 3** (Smoothness of $L$). *Let $V$ be some RKHS satisfying Assumption 1. Let $L$ be the empirical risk defined on $L^2([0,1],V)$ and associated with the RKHS-NODE model. Then there exists a continuous function $\mathbf{C} : \mathbb{R}_+ \to \mathbb{R}_+^*$ such that for every $R \geq 0$ and every $v, \bar{v} \in L^2([0,1],V)$ with $\|v\|_{L^2}, \|\bar{v}\|_{L^2} \leq R$:*

$$\|\nabla L(v) - \nabla L(\bar{v})\|_{L^2} \leq \mathbf{C}(R)\|v - \bar{v}\|_{L^2}.$$

We note $\kappa$ the constant associated with Assumption 1. The proof of Property 3 relies on the following lemma:

**Lemma 2.** *Let $v, \bar{v} \in L^2([0,1],V)$ be some control parameters and $R \geq 0$ be some radius such that $\|v\|_{L^2}, \|\bar{v}\|_{L^2} \leq R$. Let $(x,y) \in \mathbb{R}^d \times \mathbb{R}^{d'}$ be some pair of data input / objective output.*

*(i) Let $z, \bar{z}$ be solutions of Eq. (6) with parameter $v$ and $\bar{v}$ respectively and with the same initial condition $Ax$, then for any $t \in [0,1]$:*

$$\|z_t - \bar{z}_t\| \leq \kappa e^{\kappa R}\|v - \bar{v}\|_{L^2}.$$

*(ii) Let $p, \bar{p}$ be solutions of Eq. (11) with parameter $v$ and $\bar{v}$ respectively and with initial condition $\frac{1}{N}B^\top(Bz_1 - y)$ and $\frac{1}{N}B^\top(B\bar{z}_1 - y)$, then for any $t \in [0,1]$:*

$$\|p_t - \bar{p}_t\| \leq$$
$$\frac{\kappa e^{2\kappa R}\|B\|_2}{N}\|v - \bar{v}\|_{L^2}\left[\|B\|_2 + \|B(\bar{z}_1 - y)\|(1 + Re^{\kappa R})\right].$$

*Proof of Lemma 2.* **Proof of (i)** For every time $t \in [0,1]$ we have:

$$z_t - \bar{z}_t = \int_0^t \left(v_s(z_s) - \bar{v}_s(\bar{z}_s)\right)\mathrm{d}s$$
$$= \int_0^t \left(v_s(z_s) - v_s(\bar{z}_s) + v_s(\bar{z}_s) - \bar{v}_s(\bar{z}_s)\right)\mathrm{d}s,$$

and by triangle inequality:

$$\|z_t - \bar{z}_t\| \leq \int_0^t \left(\|v_s(z_s) - v_s(\bar{z}_s)\| + \|v_s(\bar{z}_s) - \bar{v}_s(\bar{z}_s)\|\right)\mathrm{d}s$$
$$\leq \int_0^t \kappa\|v_s\|_V\|z_s - \bar{z}_s\|\mathrm{d}s + \int_0^t \kappa\|v_s - \bar{v}_s\|_V\mathrm{d}s,$$

where we used Assumption 1 in the second inequality. Therefore, by Grönwall's lemma:

$$\|z_t - \bar{z}_t\| \leq \kappa e^{\kappa\|v\|_{L^2}}\int_0^t \|v_s - \bar{v}_s\|_V\mathrm{d}s$$
$$\leq \kappa e^{\kappa R}\|v - \bar{v}\|_{L^2}.$$

**Proof of (ii)** For any $t \in [0,1]$ we have:

$$p_t - \bar{p}_t = (p_1 - \bar{p}_1) - \int_1^t \left(Dv_s(z_s)^\top p_s - D\bar{v}_s(\bar{z}_s)^\top \bar{p}_s\right)\mathrm{d}s$$
$$= (p_1 - \bar{p}_1) - \int_1^t \left[Dv_s(z_s)^\top(p_s - \bar{p}_s) + \left(Dv_s(z_s) - Dv_s(\bar{z}_s)\right)^\top \bar{p}_s + \left(Dv_s(\bar{z}_s) - D\bar{v}_s(\bar{z}_s)\right)^\top \bar{p}_s\right]\mathrm{d}s,$$

and using the triangle inequality and Assumption 1:

$$\|p_t - \bar{p}_t\| \leq \|p_1 - \bar{p}_1\| + \int_t^1 \left[\kappa\|v_s\|_V\|p_s - \bar{p}_s\| + \kappa\|v_s\|_V\|z_s - \bar{z}_s\|\|\bar{p}_s\| + \kappa\|v_s - \bar{v}_s\|_V\|\bar{p}_s\|\right]\mathrm{d}s.$$

Then, using Grönwall's lemma backward in time gives:

$$\|p_t - \bar{p}_t\| \leq \|p_1 - \bar{p}_1\|e^{\kappa\|v\|_{L^2}} + \kappa e^{\kappa\|v\|_{L^2}}\int_t^1 \|v_s - \bar{v}_s\|_V\|\bar{p}_s\|\mathrm{d}s + \kappa e^{\kappa\|v\|_{L^2}}\int_t^1 \|v_s\|_V\|z_s - \bar{z}_s\|\|\bar{p}_s\|\mathrm{d}s.$$

On one hand, because of (i) we have for every $s \in [0,1]$:

$$\|z_s - \bar{z}_s\| \leq \kappa e^{\kappa R}\|v - \bar{v}\|_{L^2},$$

and also:

$$\|p_1 - \bar{p}_1\| = \frac{1}{N}\|B^\top B(z_1 - \bar{z}_1)\|$$

$$\leq \frac{\|B\|_2^2}{N}\kappa e^{\kappa R}\|v - \bar{v}\|_{L^2}.$$

On the other hand, recalling (ii) of Lemma 1, for every $s \in [0,1]$:

$$\|\bar{p}_s\| \leq \frac{\sigma_{\max}(B^\top)}{N}e^{\kappa R}\|Bz_1 - y\|.$$

Putting these estimates in the preceding inequality gives:

$$\|p_t - \bar{p}_t\| \leq \left[\frac{\|B\|_2^2}{N}\kappa e^{2\kappa R} + \frac{\sigma_{\max}(B^\top)}{N}\kappa e^{2\kappa R}\|B(\bar{z}_1 - y)\| + R\frac{\sigma_{\max}(B^\top)}{N}\kappa^2 e^{3\kappa R}\|B(\bar{z}_1 - y)\|\right]\|v - \bar{v}\|_{L^2},$$

which is the desired result.

$\square$

*Proof of Property 3.* Let $v, \bar{v} \in L^2([0,1], V)$ with $\|v\|_{L^2}, \|\bar{v}\|_{L^2} \leq R$. Then taking the same notation as in Lemma 2, we have for any $t \in [0,1]$:

$$\nabla L(v)_t - \nabla L(\bar{v})_t = \sum_{i=1}^N K(., z_t^i)p_t^i - \sum_{i=1}^N K(.\bar{z}_t^i)\bar{p}_t^i$$

$$= \sum_{i=1}^N K(., z_t^i)(p_t^i - \bar{p}_t^i) + \sum_{i=1}^N (K(., z_t^i) - K(.\bar{z}_t^i))\bar{p}_t^i,$$

and we can write $\|\nabla L(v)_t - \nabla L(\bar{v})_t\|_V \leq T_1 + T_2$ with:

$$T_1 := \|\sum_{i=1}^N K(., z_t^i)(p_t^i - \bar{p}_t^i)\|_V, \quad T_2 := \|\sum_{i=1}^N (K(., z_t^i) - K(.\bar{z}_t^i))\bar{p}_t^i\|_V.$$

First we consider deriving an upper bound on $T_1$. Note that by the definition of the norm in RKHSs and by Assumption 2 we have:

$$T_1^2 = \sum_{1 \leq i,j \leq N}(p_t^i - \bar{p}_t^i)^\top K(z_t^i, z_t^j)(p_t^j - \bar{p}_t^j) \leq \Lambda\sum_{i=1}^N \|p_t^i - \bar{p}_t^i\|^2.$$

Therefore, using (ii) from Lemma 2 to bound $\|p_t^i - \bar{p}_t^i\|$ for every index $i$ we get:

$$T_1^2 \leq \Lambda\mathbf{C}_1^2\|v - \bar{v}\|_{L^2}^2,$$

with:

$$\mathbf{C}_1^2 = \sum_{i=1}^N \frac{\kappa^2 e^{4\kappa R}\|B\|_2^2}{N^2}\left[\|B\|_2 + \|B(\bar{z}_1^i - y)\|(1 + Re^{\kappa R})\right]^2$$

$$\leq \sum_{i=1}^N \frac{2\kappa^2 e^{4\kappa R}\|B\|_2^2}{N^2}\left[\|B\|_2^2 + \|B(\bar{z}_1^i - y)\|^2(1 + Re^{\kappa R})^2\right]$$

$$\leq \frac{2\kappa^2 e^{4\kappa R}\|B\|_2^4}{N} + \frac{4\kappa^2 e^{4\kappa R}\|B\|_2^2}{N}(1 + Re^{\kappa R})^2 L(\bar{v}),$$

where we recognised $L(\bar{v})$ in the third line. By continuity of $L$ we can define for every $R \geq 0$:
$$L^*(R) := \sup_{\|v\|_{L^2} \leq R} L(v).$$
And therefore:
$$\mathbf{C}_1^2 \leq \frac{2\kappa^2 e^{4\kappa R}\|B\|_2^4}{N} + \frac{4\kappa^2 e^{4\kappa R}\|B\|_2^2}{N}(1 + Re^{\kappa R})^2 L^*(R) =: \mathbf{C}_3(R)^2.$$
We then consider deriving an upper-bound on $T_2$. By triangle inequality:
$$T_2 \leq \sum_{i=1}^{N} \|(K(.,z_t^i) - K(.,\bar{z}_t^i))\bar{p}^i{}_t\|_V.$$
Consider any $\alpha \in V$, then for any index $i \in [\![1, N]\!]$, by the reproducing property:
$$\langle (K(.,z_t^i) - K(.,\bar{z}_t^i))\bar{p}_t^i, \alpha \rangle_V = \langle \alpha(z_t^i) - \alpha(\bar{z}_t^i), \bar{p}_t^i \rangle$$
$$\leq \kappa \|\alpha\|_V \|z_t^i - \bar{z}_t^i\| \|\bar{p}_t^i\|,$$
where we used the Cauchy-Schwarz inequality and Assumption 1 applied to $\alpha$. Therefore, by duality:
$$\|(K(.,z_t^i) - K(.,\bar{z}_t^i))\bar{p}^i{}_t\|_V \leq \kappa \|z_t^i - \bar{z}_t^i\| \|\bar{p}_t^i\|.$$
Using the estimates of Lemma 1 and Lemma 2 we get:
$$\|(K(.,z_t^i) - K(.,\bar{z}_t^i))\bar{p}^i{}_t\|_V \leq \frac{\kappa^2 e^{2\kappa R}\|B\|_2}{N}\|B\bar{z}_1^i - y^i\|\|v - \bar{v}\|_{L^2}.$$
And finally, using Cauchy-Schwarz inequality and recognizing $L(\bar{v})$ we have:
$$T_2^2 \leq N \sum_{i=1}^{N} \|(K(.,z_t^i) - K(.,\bar{z}_t^i))\bar{p}^i{}_t\|_V^2$$
$$\leq \mathbf{C}_2^2 \|v - \bar{v}\|_{L^2}^2,$$
with:
$$\mathbf{C}_2^2 = 2\kappa^4 e^{4\kappa R}\|B\|_2^2 L(\bar{v})$$
$$\leq 2\kappa^4 e^{4\kappa R}\|B\|_2^2 L^*(R) =: \mathbf{C}_4(R)^2.$$
Therefore we obtain the result by setting:
$$\mathbf{C}(R) = \left[\Lambda \mathbf{C}_3(R)^2 + \mathbf{C}_4(R)^2\right]^{1/2}.$$

$\square$

Provided with Property 3, we can finish the proof of Theorem 2.

*Proof of Theorem 2.* By Property 2, $L$ satisfies the PL inqualities of Definition 2 and the proof is a direct corollary of Theorem 1. It only remains to show that the smoothness condition of Definition 3 is verified.

Let $v, \bar{v} \in L^2([0, 1], V)$ such that $\|v\|_{L^2}, \|\bar{v}\|_{L^2} \leq R$ for some radius $R \geq 0$. Then we have:
$$L(\bar{v}) = L(v) + \int_0^1 \nabla L(v + t(\bar{v} - v)).(\bar{v} - v)\mathrm{d}t$$
$$= L(v) + \nabla L(v).(\bar{v} - v)$$
$$+ \int_0^1 \left[\nabla L(v + t(\bar{v} - v)) - \nabla L(v)\right] \cdot (\bar{v} - v)\mathrm{d}t.$$
Using Property 3, there exists some $\mathbf{C}(R)$ such that:
$$\|\nabla L(v + t(\bar{v} - v)) - \nabla L(v)\|_{L^2} \leq t\mathbf{C}(R)\|\bar{v} - v\|_{L^2}.$$
This gives the inequality:
$$L(\bar{v}) \leq L(v) + \nabla L(v) \cdot (\bar{v} - v) + \frac{\mathbf{C}(R)}{2}\|\bar{v} - v\|_{L^2}^2,$$
which is the desired result.

$\square$

# D   Proofs of Section 5

The results in Section 5 show how the condition for convergence in Eq. (15) can be enforced by considering suitable RKHSs of vector-fields and suitable matrices $A$ and $B$. We give in Appendix D.3 examples of suitable kernels.

In the following, we assume that for every $q \geq 1$ we are provided with a function $k_q : \mathbb{R}_+ \to \mathbb{R}$ such that the induced symmetric rotationally-invariant kernel $K_q$ defined by:

$$\forall z, z' \in \mathbb{R}^q, \ K_q(z, z') = k_q(\|z - z'\|) \operatorname{Id}_q, \tag{23}$$

is a positive-definite kernel over $\mathbb{R}^q$. Without loss of generality, one can assume $k_q$ to be normalized, that is $k_q(0) = 1$. We note $V_q$ the vector-valued RKHS associated with $K_q$. The properties of $V_q$ are then entirely determined by $k_q$. In particular, smoothness of the kernel at $0$ implies regularity of the vector-fields in $V_q$:

**Property 4** (Regularity of $V_q$). *Let $k_q : \mathbb{R}_+ \to \mathbb{R}$ be some function defining a positive symmetric kernel $K_q$. If $k_q$ is $4$ times differentiable at $0$, with $k_q'(0) = k_q^{(3)}(0) = 0$. Then $V_q$ satisfies Assumption 1 with constant $\kappa = \sqrt{k_q(0)} + \sqrt{-k_q''(0)} + \sqrt{k_q^{(4)}(0)}$.*

As a consequence, if the derivatives of $k_q$ can be bounded uniformly over $q$ then $V_q$ satisfies Assumption 1 with some constant $\kappa$ independent of $q$. This, is the case for the Matérn kernel $k$ defined in Eq. (17).

*Proof.* The proof proceeds by duality arguments. For $q \geq 1$, consider some $v \in V_q$. Then for any $z \in \mathbb{R}^q$ and any $\alpha \in V_q$, by the reproducing properties of RKHSs:

$$
\begin{aligned}
\langle v(z), \alpha \rangle &= \langle v, K_q(., z)\alpha \rangle_{V_q} \\
&\leq \|v\|_{V_q} \|K_q(., z)\alpha\|_{V_q} \\
&= \|v\|_{V_q} \big(\langle \alpha, K_q(z, z)\alpha \rangle\big)^{1/2} \\
&\leq \sqrt{k_q(0)} \|v\|_{V_q} \|\alpha\|.
\end{aligned}
$$

Therefore, by duality $\|v(z)\| \leq \sqrt{k_q(0)} \|v\|_{V_q}$ and then by taking the supremum over $z \in \mathbb{R}^q$:

$$\|v\|_\infty \leq k_q(0)\|v\|_{V_q}.$$

Then for any $z \in \mathbb{R}^q$ any $\alpha, \beta \in \mathbb{R}^q$ and any $h \in \mathbb{R}_+$:

$$
\begin{aligned}
&\langle v(z + h\alpha) - v(z), \beta \rangle \\
&= \langle v, (K_q(., z + h\alpha) - K_q(., z))\beta \rangle \\
&\leq \|v\|_{V_q} \|(K_q(., z + h\alpha) - K_q(., z))\beta\|_{V_q}.
\end{aligned}
$$

In the r.h.s we have using Taylor's expansion of $k_q$ at $0$:

$$
\begin{aligned}
\|(K_q(., z + h\alpha) - K_q(., z))\beta\|_{V_q}^2 &= \begin{pmatrix} \beta \\ -\beta \end{pmatrix}^\top \begin{pmatrix} k_q(0)Id_q & k_q(h\|\alpha\|)Id_q \\ k_q(h\|\alpha\|)Id_q & k(0)Id_q \end{pmatrix} \begin{pmatrix} \beta \\ -\beta \end{pmatrix} \\
&= 2\|\beta\|^2 (k_q(0) - k_q(h\|\alpha\|)) \\
&= -\|\beta\|^2 h^2 \|\alpha\|^2 k_q''(0) + o(h^2).
\end{aligned}
$$

Taking the limit $h \to 0$:

$$
\begin{aligned}
\langle Dv(z)\alpha, \beta \rangle &= \lim_{h \to 0} h^{-1} \langle v(z + h\alpha) - v(z), \beta \rangle \\
&\leq \sqrt{-k_q''(0)} \|v\|_{V_q} \|\alpha\| \|\beta\|,
\end{aligned}
$$

and therefore $\|Dv(z)\|_2 \leq \sqrt{-k_q''(0)} \|v\|_{V_q}$.

Finally, let us bound $\|D^2 v\|_{2,\infty}$. For any $z \in \mathbb{R}^q$ any $\alpha, \beta, \gamma \in \mathbb{R}^q$ and any $h, l \geq 0$ we have in the same manner:

$$\langle v(z + h\beta + l\alpha) - v(z + h\beta) - v(z + l\alpha) + v(z), \gamma \rangle$$

$$\leq \|v\|_{V_q} \|\beta\| \|\alpha\| \|\gamma\| hl \sqrt{k_q^{(4)}(0)} + o(hl)$$

where the second line is obtained by Taylor expansion of $k_q$ at 0. Thus, taking the limit $h, l \to 0$:

$$\langle D^2 v(z)(\alpha, \beta), \gamma \rangle = \lim_{h,l \to 0} h^{-1} l^{-1} \langle v(z + h\beta + l\alpha) - v(z + h\beta) - v(z + l\alpha) + v(z), \gamma \rangle$$

$$\leq \sqrt{k_q^{(4)}(0)} \|v\|_{V_q} \|\beta\| \|\alpha\| \|\gamma\|,$$

and therefore $\|D^2 v(z)\|_2 \leq \sqrt{k_q^{(4)}(0)} \|v\|_{V_q}$.

Setting $\kappa = \sqrt{k_q(0)} + \sqrt{-k_q''(0)} + \sqrt{k_q^{(4)}(0)}$ we obtain the result. Moreover, choosing appropriate $v$ in the above proof, inequalities become sharp and one observes that the constant $\kappa$ is optimal.

$\square$

## D.1 Enforcing convergence with high dimensional lifting and universal kernels

Here we investigate the dependency of Eq. (15) w.r.t. $q$, $\delta$ and $N$ for the class of RKHS $V_q$ and thereby recover the proof of Proposition 1.

We make the following assumption concerning the decay of $k_q$ at infinity:

**Assumption 3** (Decay of $k_q$). *For every $q \geq 1$, $k_q(x)$ tends to 0 when $x$ tends to infinity and we note $\beta_{q,N} > 0$ s.t.:*

$$\forall x \geq \beta_{q,N}, \ |k_q(x)| \leq \frac{1}{2N}.$$

*Moreover for fixed $N$ we assume that*

$$\beta_{q,N} = o_{q \to +\infty}(q^{1/4}).$$

### D.1.1 Lifting matrices

For any $q \geq 1$ we consider here the matrices:

$$A_q := q^{-1/4}(\mathrm{Id}_d, ..., \mathrm{Id}_d, 0)^\top \in \mathbb{R}^{q \times d},$$

$$B_q := q^{1/4}(\mathrm{Id}_{d'}, 0...0) \in \mathbb{R}^{d' \times q},$$

where there are $\lfloor q/d \rfloor$ copies of $\mathrm{Id}_d$ in $A_q$. In particular we have:

$$\sigma_{\min}(A_q) = q^{-1/4}\sqrt{\lfloor q/d \rfloor} \simeq q^{1/4},$$

$$\sigma_{\min}(B_q^\top) = \sigma_{\max}(B_q^\top) = q^{1/4}$$

and $B_q A_q \in \mathbb{R}^{d' \times d}$ is independent of $q$. We also consider for every $q \geq 1$ some control parameter initialization $V_q^0 \in L^2(V_q)$ such that $\|v_q^0\|_{L^2} \leq R_0 q^{-1/4}$ and assume the data distribution to be compactly supported.

**Proposition 4.** *Let $R > 0$ and $d, d' \geq 1$. Assume Assumption 3 is satisfied, $V_q$ satisfies Assumption 1 with constant $\kappa$ independent of $q$ and there exists $R_0 > 0$ s.t. $\|v_q^0\| \leq R_0 q^{-1/4}$ for every $q \geq 1$.*
*Then there exists some constant $C > 0$ so that for any $N \geq 2$ and any $\delta \in (0, 1]$, Eq. (15) is satisfied with matrices $A_q$, $B_q$ and $\kappa, \lambda, \Lambda$ associated with the RKHS $V_q$ as soon as:*

$$q \geq CN^4, \ and \ q \geq C\delta^{-4}\beta_{q,N}^4. \tag{24}$$

Note that the second condition in Eq. (24) can always be ensured for large enough $q$ thanks to Assumption 3. In the case of the Matérn kernel $k$ defined in Eq. (17), such an assumption is verified because it has exponential decay and it is independent of $q$. Hence, Proposition 1 is a direct consequence of Proposition 4.

*Proof of Proposition 4.* Let $q \geq 1$. Using the fact that $d^2 \lfloor q/d \rfloor^2 \geq q(q - 2d)$, considering:

$$q \geq 2d + d^2 \frac{\beta_{q,N}^4}{\delta^4 e^{-4\kappa(R+R_0)}} \tag{25}$$

is enough to ensure that:

$$q^{-1/4} \sqrt{\lfloor q/d \rfloor} \delta e^{-\kappa(R+R_0)} \geq \beta_{q,N}.$$

Then, by Assumption 3 for $(z^i)_{1 \leq i \leq N} \in (\mathbb{R}^q)^N$ with data separation $q^{-1/4} \sqrt{\lfloor q/d \rfloor} \delta e^{-\kappa(R+R_0)}$ we have:

$$\forall 1 \leq i < j \leq N, \ |k_q(\|z^i - z^j\|)| \leq \frac{1}{2N}.$$

Thus, the kernel matrix $\mathbb{K} = (k_q(\|z^i - z^j\|) \operatorname{Id}_q)_{i,j}$ is diagonally dominant with:

$$\lambda_{\min}(\mathbb{K}) \geq 1 - \frac{N-1}{2N} \geq \frac{1}{2},$$

and by definition of $\lambda$ in Eq. (15):

$$\lambda(\sigma_{\min}(A_q)^{-1} \delta^{-1} e^{\kappa(R+R_0)}) \geq \frac{1}{2}. \tag{26}$$

Moreover, $\Lambda \leq N$ because $k_q$ is bounded by 1.

Let $x \in B(0, r_0)$ and assume $z$ is a solution of Eq. (6) for the control parameter $v_q^0$ and with initial condition $A_q x$. We have at time $t = 1$:

$$z_1 = A_q x + \int_0^1 (v_q^0)_t(z_t) \mathrm{d}t,$$

so that by triangle inequality and Assumption 1:

$$\|z_1 - A_q x\| \leq \kappa \|v_q^0\|_{L^2},$$

and then because $\|v_q^0\| \leq R_0 q^{-1/4}$ and the dataset is compactly supported:

$$\begin{aligned}
\|F(v_q^0, x)\| = \|B_q z_1\| \\
\leq \|B_q A_q x\| + \|B_q(z_1 - A_q x)\| \\
\leq \|B_q A_q\|_2 r_0 + \kappa R_0,
\end{aligned}$$

with $B_q A_q$ independent of $q$. Thus $L(v_q^0) \leq C$ for some constant $C$ independent of $q$, $N$ and $\delta$.

Finally:

$$\frac{\sigma_{\max}(B_q^\top)}{\sigma_{\min}(B_q^\top)^2} = q^{-1/4}, \tag{27}$$

and putting Eq. (26) and Eq. (27) into the l.h.s. Eq. (15) gives:

$$\frac{2\sqrt{2}\sigma_{\max}(B_q^\top)\sqrt{N\Lambda L(0)} e^{3\kappa(R+R_0)}}{\sigma_{\min}(B_q^\top)^2 \lambda(\sigma_{\min}(A_q)^{-1}\delta^{-1}e^{-\kappa(R+R_0)})} \leq 4\sqrt{2C} e^{3\kappa(R+R_0)} \frac{N}{q^{1/4}}.$$

Considering $R > 0$ is fixed (c.f. Remark 7), Theorem 2 can be applied as soon as:

$$q \geq 2^{10} C^2 e^{12\kappa(R+R_0)} R^{-4} N^4 \tag{28}$$

and combining this bound with the one in Eq. (25) gives the result. $\qquad \square$

**Remark 7** (Choice of $R$). *The proof of Proposition 4 holds for any fixed $R > 0$ whose choice impacts the result through the constant $C$. There is a trade-off between minimizing $e^{4\kappa R}$ to have a better dependency of $q$ w.r.t. $\delta^{-1} \log(N)$ in Eq. (25) and minimizing $R^{-1} e^{3\kappa R}$ to have a better dependency w.r.t. $N$ in Eq. (28). However, in any case, optimizing w.r.t. $R$ only improves the result up to a constant factor.*

### D.1.2 Scaling matrices

For $\alpha > 0$, we consider here the matrices:

$$A = \alpha(\mathrm{Id}_d, 0)^\top \in \mathbb{R}^{(d+d')\times d} \quad \text{and} \quad B = \alpha(0, \mathrm{Id}_{d'}) \in \mathbb{R}^{d' \times (d+d')}.$$

Then, in the proof of Proposition 4 one has $\sigma_{\min}(A) = \alpha$ and thus Eq. (26) holds as soon as:

$$\alpha \geq \delta^{-1} e^{\kappa(R+R_0)} \beta_{d+d',N}.$$

Moreover, $\sigma_{\max}(B^\top) = \sigma_{\min}(B^\top) = \alpha$ and $F(0, x) = 0$ for every input $x$ as $BA = 0$. Thus, with initialization $v^0 = 0$ the l.h.s. of Eq. (15) scales as:

$$\frac{2\sqrt{2}\sigma_{\max}(B^\top)\sqrt{N\Lambda L(0)}e^{3\kappa R}}{\sigma_{\min}(B^\top)^2 \lambda(\sigma_{\min}(A)^{-1}\delta^{-1}e^{-\kappa R})} \leq 4\sqrt{2C}e^{3\kappa R}\frac{N}{\alpha} = \mathcal{O}(1/\alpha),$$

and global convergence holds for $\alpha = \Omega(\delta^{-1}\beta_{d+d',N} + N)$.

### D.2 Enforcing convergence with high dimensional embedding en finite dimensional kernels

We recover here the result of Proposition 2 for the more general kernel $k_q$. In particular notice that, as an application of Bochner's theorem [50], for every $q \geq 1$ there exists some probability measure $\mu_q$ over $\mathbb{R}^q$ such that:

$$\forall z \in \mathbb{R}^q, \ k_q(\|z\|) = \int_{\mathbb{R}^q} e^{i\langle z, \omega\rangle} \mathrm{d}\mu_q(\omega). \tag{29}$$

Then, such as in Eq. (20) for the Matérn kernel, for any independent sampling $\omega^j \sim \mu_q$ of size $r$ one can consider the feature map:

$$\varphi : z \mapsto \left(e^{i\langle z, \omega^j\rangle}\right)_{1 \leq j \leq r} \in \mathbb{C}^r. \tag{30}$$

Such a feature map induces a structure of RKHS $\hat{V}_q$ which is the set of residuals of Eq. (3) with activation $\varphi$. The associated kernel is $\hat{K}_q : (z, z') \mapsto \hat{k}_q(z, z') \, \mathrm{Id}_q$ with:

$$\forall z, z' \in \mathbb{R}^q, \ \hat{k}_q(z, z') := \langle \varphi(z), \varphi(z')\rangle$$
$$\xrightarrow{r \to +\infty} k_q(\|z - z'\|),$$

almost surely, by the law of large numbers.

We make the following assumption on $\mu_q$:

**Assumption 4** (Moments of $\mu_q$). *The measure $\mu_q$ admits finite moments up to order 8:*

$$\mathbb{E}_{\mu_q}\left[\prod_{j=1}^{8} |\omega_{i_j}|\right] < \infty, \ \forall i_1, ..., i_8 \in [\![1, q]\!].$$

*Moreover, we assume those moments are independent of $q$.*

Note that Assumption 4 implies regularity on the function $k_q$. Indeed by Fourier inversion theorem we have for every $r \in \mathbb{R}_+$ and every $\theta \in \mathbb{S}^{d-1}$:

$$k_q(r) = \mathbb{E}_{\mu_q}\left[e^{ir\langle\theta, \omega\rangle}\right].$$

By theorems of derivation under the integral $k_q$ is $8^{th}$-time differentiable on $\mathbb{R}_+$ and for $0 \leq l \leq 8$:

$$k_q^{(l)}(r) = \mathbb{E}_{\mu_q}\left[(i\langle\theta, \omega\rangle)^l e^{ir\langle\theta, \omega\rangle}\right].$$

In particular, $k_q$ is four time differentiable at 0 and:

$$k'(0) = \mathbb{E}_{\mu_q}\left[i\langle\theta, \omega\rangle\right]$$
$$k^{(3)}(0) = \mathbb{E}_{\mu_q}\left[-i\langle\theta, \omega\rangle^3\right]$$

Therefore, $k_q'(0)$ and $k_q^{(3)}(0)$ are in $i\mathbb{R} \cap \mathbb{R} = \{0\}$ and Property 4 holds. Moreover, as the moments are independent of $q$, the associated $\kappa$ is also independent of $q$.

**Proposition 5.** *Consider $q, N \geq 1$ and $\epsilon, \tau, R > 0$.*

*(i) Assume Assumption 4 is satisfied. For $r \geq \Omega(\tau q^8)$, with probability greater than $1 - \tau^{-1}$, $\hat{V}_q$ satisfies Assumption 1 with some $\hat{\kappa} \leq \kappa + 1$.*

*(ii) For $r \geq \Omega(\epsilon^{-2} N^2 (q \log(\|A\|_2 r_0 + R) + \tau))$, with probability greater than $1 - e^{-\tau}$, for any control parameter $v \in L^2([0,1], \hat{V}_q)$ s.t. $\|v\|_{L^2} \leq R$ and any time $t \in [0,1]$:*

$$\lambda_{\min}(\hat{\mathbb{K}}((z_t^i)_i)) \geq \lambda_{\min}(\mathbb{K}((z_t^i)_i)) - \epsilon,$$

*where the $(z^i)_i$ are the solutions to Eq. (6) and $\hat{\mathbb{K}}$, $\mathbb{K}$ are the kernel matrices associated with $\hat{k}$ and $k$ respectively.*

As Assumption 4 is satisfied for the Matérn kernel $k$ defined in Eq. (17) as soon as $\nu > 4$, Proposition 2 is a direct consequence of Proposition 5.

*Proof of Proposition 5.* **Proof of (i)** We already saw that thanks to the assumption on the moments of $\mu_q$, the RKHS $V_q$ associated with $k_q$ satisfies Assumption 1 with constant $\kappa$.

Then we want to prove that for sufficiently high $r$, the RKHS $\hat{V}_q$ generated by the feature map $\varphi$ in Eq. (20), satisfies Assumption 1.

Let $v \in \hat{V}_q$ be of the form:

$$v : z \mapsto W\varphi(z)$$

for some $W \in \mathbb{R}^{q \times r}$. For $z \in \mathbb{R}^q$, $\|\varphi(z)\| = 1$ and thus:

$$\|v(z)\| = \|W\varphi(z)\| \leq \|W\| = \|v\|_{\hat{V}_q},$$

so that $\|v\|_\infty \leq \|v\|_{\hat{V}_q}$.

Then $Dv(z) = WD\varphi(z)$ and by the law of large number we have for any $\theta \in \mathbb{S}^{q-1}$:

$$\|D\varphi(z)\theta\|^2 = \frac{1}{r} \sum_{j=1}^r \sum_{1 \leq k,l \leq q} \omega_k^j \omega_l^j \theta_k \theta_l$$

$$= \frac{1}{r} \sum_{j=1}^r \langle \omega^j, \theta \rangle^2$$

$$\xrightarrow{r \to +\infty} \mathbb{E}_{\mu_q}\left[\langle \omega, \theta \rangle^2\right] = -k_q''(0).$$

Because $\mu_q$ admits finite fourth order moments, the rate of convergence can be controlled using Chebyshev's inequality. For every indices $k, l \in [\![1, q]\!]$:

$$\mathbb{P}\left(\left|\frac{1}{r} \sum_{j=1}^r \omega_k^j \omega_l^j - \mathbb{E}_{\mu_q}[\omega_k \omega_l]\right| \geq \alpha/q\right) \leq \frac{q^2 \mathbb{E}_{\mu_q}[\omega_k^2 \omega_l^2]}{\alpha^2 r}.$$

For $r \geq \Omega(\frac{q^4 \tau}{\alpha^2})$ we have with probability greater than $1 - \tau^{-1}$ that the above inequality is satisfied for every indices $k, l$. Thus for every $z \in \mathbb{R}^q$ and every $\theta \in \mathbb{S}^{q-1}$:

$$\left|\|D\varphi(z)\theta\|^2 + k_q''(0)\right| \leq \sum_{1 \leq k,l \leq q} |\theta_k \theta_l| \left|\frac{1}{r} \sum_{j=1}^r \omega_k^j \omega_l^j - \mathbb{E}_{\mu_q}[\omega_k \omega_l]\right|$$

$$\leq \sum_{1 \leq k,l \leq q} |\theta_k \theta_l| \frac{\alpha}{q}$$

$$\leq \alpha,$$

using Chauchy-Schwarz inequality in the last line. We can thus conclude:

$$\|D\varphi\|_{2,\infty}^2 \leq -k_q''(0) + \alpha.$$

The same arguments holds for $D^2 v(z) = W D^2 \varphi(z)$. For any $\theta \in \mathbb{S}^{q-1}$ we have:

$$D^2\varphi(z)(\theta,\theta) = \left(\frac{1}{\sqrt{r}} \sum_{1\le k,l\le q} -e^{\imath\langle z, \omega^j\rangle} \omega_k^j \omega_l^j \theta_k \theta_l\right)_{1\le j\le r}.$$

Passing to the squared norm we get:

$$\|D^2\varphi(z)(\theta,\theta)\|^2 = \frac{1}{r}\sum_{j=1}^r \sum_{1\le k,l,s,t\le q} \omega_k^j \omega_l^j \omega_s^j \omega_t^j \theta_k\theta_l\theta_s\theta_t$$
$$\xrightarrow{r\to+\infty} \sum_{1\le k,l,s,t\le q} \mathbb{E}_{\mu_q}\left[\omega_k\omega_l\omega_s\omega_t\right]\theta_k\theta_l\theta_s\theta_t$$
$$= \mathbb{E}_{\mu_q}\left[\langle\omega,\theta\rangle^4\right] = k_q^{(4)}(0).$$

Then because $\mu_q$ admits $8^{th}$ order moments, we can control the convergence in probability by Chebyshev's inequality. For $r \ge \Omega(\frac{q^8\tau}{\alpha^2})$ we have with probability greater than $1 - \tau^{-1}$:

$$\|D^2\varphi\|_{2,\infty}^2 \le k_q^{(4)}(0) + \alpha.$$

Finally $\hat{V}_q$ satisfies Assumption 1 with:

$$\hat{\kappa} \le (k_q(0))^{1/2} + (-k_q''(0))^{1/2} + (k_q^{(4)}(0))^{1/2} + 1$$

for $\alpha$ sufficiently low.

**Proof of (ii).** For $t \in [0,1]$, we consider $(z_t^i)_i$ the solutions of of Eq. (6) for some control parameter $v \in L^2([0,1], \hat{V}_q)$ and we introduce the kernel matrices:

$$\hat{\mathbb{K}}_t = (\hat{K}_q(z_t^i, z_t^j))_{1\le i,j\le N}, \quad \mathbb{K}_t = (K_q(z_t^i, z_t^j))_{1\le i,j\le N}.$$

Using the first point, we know that if $\|v\|_{L^2} \le R$, then $\|z_t^i\| \le \|Ax^i\| + (\kappa+1)R$. Then, using Theorem 1 in [53], we have for every indices $i,j$ and every $t \in [0,1]$:

$$\mathbb{P}\left(|\hat{k}(z_t^i, z_t^j) - k(\|z_t^i - z_t^j\|)| \ge \frac{h(q,R) + \sqrt{2\tau}}{\sqrt{r}}\right) \le e^{-\tau},$$

with $h(q,R) := \mathcal{O}(\sqrt{q\log(\|A\|_2 r_0 + R)})$. Thus, choosing $r \ge \Omega\left(\epsilon^{-2}N^2(q\log(\|A\|_2 r_0 + R) + \tau)\right)$, we have with probability greater than $1 - e^{-\tau}$, $\lambda_{\min}(\hat{\mathbb{K}}_t) \ge \lambda_{\min}(\mathbb{K}_t) - \epsilon$, for any $t \in [0,1]$.

$\square$

Note that the assumption of finite $8^{th}$ moments is only needed to control the convergence rate of $\hat{k}_q$ towards $k_q$ in probability. By the law of large numbers, assuming finite $4^{th}$-order moments is sufficient to have convergence almost surely. Also, we used the Chebyshev's inequality in order to control the convergence rate. Making stronger assumptions on the decay of $\mu_q$ (e.g. sub-gaussianity) could have led to faster convergence by using sharper concentration inequalities.

### D.3 Example of appropriate kernels

We show here that the Matérn kernel of parameter $\nu \in (8, +\infty]$ satisfies Assumption 3 and Assumption 4.

**Gaussian kernel.** The Gaussian kernel defined by for some parameter $\sigma > 0$ by $k_q(r) = e^{-\frac{\sigma^2 r^2}{2}}$. In this case the frequency distribution $\mu_q$ is the multivariate normal of variance $\sigma$ and has a density given for every $\omega \in \mathbb{R}^q$ by:

$$\mu_q(\omega) = \frac{1}{(2\pi\sigma^2)^{q/2}} e^{-\frac{\|\omega\|^2}{2\sigma^2}},$$

This distribution admits finite moments of every order which are independent of $q$. Also, $k_q$ is four times differentiable at 0 and by Property 4 the associated $V_q$ is (strongly) admissible with $\kappa = 2+\sqrt{3}$

Moreover Assumption 3 as one has $|k_q(x)| \leq 1/2N$ if:

$$x \geq \beta_{q,N} = \frac{2}{\sigma^2}\sqrt{\log(2N)}.$$

**Matérn kernel.** Sobolev spaces $H^s(\mathbb{R}^q, \mathbb{R}^q)$ are RKHSs as soon as $s > q/2$. Given some $\nu > 0$, the kernel $k_q$ associated with $H^{(q/2+\nu)}(\mathbb{R}^q, \mathbb{R}^q)$ is independent of $q$ and is defined in Eq. (17). It is associated with the multivariate t-distribution:

$$\mu_q(\omega) = C(q,\nu)(1 + \frac{\|\omega\|^2}{2\nu})^{-(\nu+q/2)},$$

for some normalising constant $C(q,\nu)$. Therefore, $\mu_q$ admits $l^{th}$ order moments as soon as $\nu \geq l/2$, and those moments are bounded independently of $q$ (see [28] for the computation of moments). In particular, for $\nu > 2$, $k_q$ is four times differentiable at 0 with $k''(0) = \nu/(\nu - 1)$ and $k^{(4)}(0) = 3\nu^2/(\nu - 1)(\nu - 2)$. Thus by Property 4, $V_q$ is (strongly) admissible with:

$$\kappa = 1 + \sqrt{\frac{\nu}{(\nu - 1)}} + \sqrt{\frac{3\nu^2}{(\nu - 1)(\nu - 2)}}.$$

Because $k_q$ has exponential decay (see [31]), there exist constants $H_\nu, G_\nu$ such that:

$$|k_q(r)| \leq G_\nu e^{-H_\nu^{-1}r}$$

and Assumption 3 is satisfied with

$$\beta_{q,N} = H_\nu \log(2G_\nu N).$$

**Remark 8** (Sampling). *Sampling over $\mu_q$ can be achieved using that for $Y \sim \mathcal{N}(0, \mathrm{Id}_q)$ and for $u$ distributed according to $\chi^2_{2\nu}$, the chi-squared distribution with $2\nu$ degrees of freedom, $Y/\sqrt{u/2\nu}$ is distributed according to $\mu_q$.*

## E RKHS-NODE as a generalization of linear networks

In an attempt to better understand the convergence properties of GD in the training of ResNets, lots of attention has first been brought towards the study of linear models, for which the training dynamic is now well understood [24, 7, 64]. We explain here in what extent our work can be seen, at least formally, as a generalization of these results to a more general class of ResNets. In this purpose, we highlight the similarity between Theorem 2, which applies to the whole class of models described by Definition 1, and [64, Theorem 3.1.], which only applies to linear ResNets.

More precisely, [64] studies model of the form:

$$F(W, x) := B(\mathrm{Id} + \frac{1}{D}W_D)...(\mathrm{Id} + \frac{1}{D}W_1)Ax, \tag{31}$$

where $x \in \mathbb{R}^d$ is the input data, $W = (W_1, ..., W_D) \in (\mathbb{R}^{q \times q})^D$ is the trained parameter and $A \in \mathbb{R}^{q \times d}$, $B \in \mathbb{R}^{d' \times q}$ are fixed matrices. Taking the limit of infinite depth $D \to +\infty$ in the above model motivates the following definition for linear Neural ODE models:

**Definition 4** (Linear-NODE). *Let $A \in \mathbb{R}^{q \times d}$ and $B \in \mathbb{R}^{d' \times q}$ be fixed matrices. Then for $W \in L^2([0,1], \mathbb{R}^{q \times q})$ and input $x \in \mathbb{R}^d$, the Linear-NODE output is given by $F(W, x) := BU_1Ax$, where $U$ is the solution to the following forward problem:*

$$\dot{U}_t = W_t U_t, \quad and \quad U_0 = \mathrm{Id}_{\mathbb{R}^q}.$$

One sees that the ResNet $F$ has residual terms that are linear w.r.t. the parameters and thus fits in the framework of our analysis. More precisely, the Linear-NODE of Definition 4 can be seen as a special instance of RKHS-NODE of Definition 1 with space of residual defined as:

$$V := \{v : z \mapsto Wz, \ W \in \mathbb{R}^{q \times q}\}.$$

This corresponds to Eq. (3) with the choice of feature map $\varphi = \mathrm{Id} : \mathbb{R}^q \to \mathbb{R}^q$. The set of residuals $V$ is then of course a RKHS for the Frobenius metric on matrices. In particular $V$ satisfies an analog of Assumption 1 in the sense that for $(v : z \mapsto Wz) \in V$:

$$\max\{ \sup_{\|z\|=1} \|v(z)\|, \ \sup_{\|z\|=1} \|Dv(z)\|, \ \sup_{\|z\|=1} \|D^2 v(z)\| \} \leq \|W\| = \|v\|_V.$$

Universality (Assumption 2) is also satisfied on full-rank data matrices. If $Z = (z^1|...|z^N) \in \mathbb{R}^{q \times N}$ then the associated kernel matrix verifies:

$$\lambda_{\min}(\mathbb{K}((z^i))) = \lambda_{\min}(Z^\top Z) = \sigma_{\min}(Z)^2,$$
$$\lambda_{\max}(\mathbb{K}((z^i))) = \lambda_{\max}(Z^\top Z) = \sigma_{\max}(Z)^2.$$

As in our above presentation we consider training Linear-NODE for the minimization of the empirical risk associated to the square euclidean distance on the output space $\mathbb{R}^{d'}$. Given data matrices $X = (x^1|...|x^N) \in \mathbb{R}^{d \times N}$ for the input and $Y = (y^1|...|y^N) \in \mathbb{R}^{d' \times N}$ for the output, we aim at finding a control parameter minimizing the risk defined for every $W \in L^2([0,1], \mathbb{R}^{q \times q})$ as:

$$L(W) := \frac{1}{2N} \sum_{i=1}^{N} \|F(W, x^i) - y^i\| = \frac{1}{2N}\|BU_1 AX - Y\|^2.$$

One difference with the previous analysis is that one can not expect the empirical risk to reach the value $0$ if the target data $Y$ is not in the linear span of the input $X$. We are thus interested in minimizing the excess risk defined as:

$$\tilde{L}(W) := L(W) - L^*$$

with $L^* := \inf_{U \in \mathbb{R}^{q \times q}} \frac{1}{2N}\|BUAX - Y\|^2$.

Following the line of the proof of Property 2, one can then show that the excess risk $\tilde{L}$ associated to our Linear-NODE model verifies the following (local) PL property:

$$\forall W \in L^2([0,1], \mathbb{R}^{q \times q}), \quad 2m(\|W\|)\tilde{L}(W) \leq \|\nabla \tilde{L}(W)\|^2 \leq 2M(\|\tilde{W}\|)\tilde{L}(W),$$

where $m$ and $M$ are given for $R \geq 0$ by:

$$m(R) = \frac{1}{N}\sigma_{\min}(B^\top)^2 \sigma_{\min}(A)^2 \sigma_r(X)^2 e^{-2R}, \quad M(R) = \frac{1}{N}\sigma_{\max}(B^\top)^2 \sigma_{\max}(A)^2 \sigma_{\max}(X)^2 e^{2R},$$

with $\sigma_r(X)$ the smallest positive singular value of $X$. Hence, in the same way local PL implies local convergence for a general RKHS $V$ (Theorem 2), convergence in the linear case follows as an application of Theorem 1:

**Theorem 4** (analog to Theorem 3.1. in [64]). *Let $W_0$ be some control parameter initialization with norm $\|W_0\| = R_0$ and assume there exists some $R > 0$ s.t.:*

$$\sqrt{8} \frac{\sigma_{\max}(B^\top)\sigma_{\max}(A)\sigma_{\max}(X)}{\sigma_{\min}(B^\top)^2 \sigma_{\min}(A)^2 \sigma_r(X)^2} \sqrt{L(W_0) - L^*} \leq Re^{-3(R+R_0)}$$

*then, for a sufficiently small step-size $\eta$, GD initialized at $W_0$ converges towards a global minimizer of $L$ with linear convergence rate.*