# OpenReview forum: "On global convergence of ResNets: From finite to infinite width using linear parameterization"
_NeurIPS.cc/2022/Conference — NeurIPS 2022 Accept_

### Official Review · Reviewer_jSRi · 2022-07-04

**Rating:** 7
**Confidence:** 5
**Soundness:** 4 excellent
**Presentation:** 3 good
**Contribution:** 4 excellent

**Summary:**

This paper provided a local Polyak-Lojasiewicz inequality for a kernelized neural ODE-model. This paper's analysis is quite interesting, but still relationship between the proposed work and the previous should be improved.

**Questions:**

I guess the authors need to discuss the realtionship between [1], which provided the convergence properties of Gradient Descent for linear ResNets. Can your analysis be considered as iinfiinte dimension analysis of [1]?
[1] M. HARDT AND T. MA, Identity Matters in Deep Learning, arXiv:1611.04231 [cs, stat] (2018). arXiv: 1611.04231.

What's the relationship between your analysis and the Resnet's NTK (which can be written down as an ODE). What's the further degree of freedom increased in your paper. At the same time, this paper is missing reference [2]. I'm curious about the relationship between your paper with [2]. Can the analysis in [2] be considered as a PL analysis?

[2] Ding Z, Chen S, Li Q, et al. On the Global Convergence of Gradient Descent for multi-layer ResNets in the mean-field regime[J]. arXiv preprint arXiv:2110.02926, 2021.

The assumption 2 is the core assumption to imply the PL condition. Can the analysis in line 256-258 be more concrete?  In there a principled way to construct such RKHS?

**Limitations:**

Thi limitation of this paper is two fold in my mind:
- i don't know how strong is assumption 2
- the hidden weight is not trained. [ but i think this model is still more powerful than NTK model.

**Strengths And Weaknesses:**

This paper provided a local Polyak-Lojasiewicz inequality for a kernelized neural ODE-model. This paper's analysis is quite interesting, but still the relationship between the proposed work and the previous should be improved.


This paper present a nice Polyak-Lojasiewicz property for the Neural ODE model. I think this paper definitely will make impact in this area. I'm willing to upgrade my score if the paper can make the relationship between the proposed work and the previous more clear.  (detailed points see  Questions)

---

> ### Author Response · Authors · 2022-08-02
> **Can your analysis be considered as infinite dimension analysis of "Identity matters in Deep Learning" (Hardt and Ma, 2018) ?**
>
> Indeed (up to some minor adaptations explained below), our analysis contains as special case deep linear ResNet and can therefore be seen as a non-linear and possibly infinite dimensional generalization. The key difference is that the ResNet structure we treat encode non-linear maps which is much closer to practical cases. Indeed, beyond linear networks which were important first steps in the theoretical understanding, we believe it is an important feature of our work to provide an analysis structure that easily adapts to different non-linear (or linear) models or data structure.
>
> As shown below, one can for example recover the convergence result of "On the Global Convergence of training deep linear ResNets" (Zou et al. 2020). We will add these details in the supplementary materials of the paper. In the case of linear ResNets one is concerned with the following space of residuals:
> $$
>     V = \lbrace v: z \mapsto W z, \ W \in \mathbb{R}^{q \times q} \rbrace ,
> $$
>
> which is a RKHS for the metric $\langle v, v' \rangle_V = \langle W, W' \rangle_F$ where we make the identification $v \leftrightarrow W$ and $\langle ., . \rangle_F$ is the Frobenius metric on matrices. This corresponds to eq. (3) with feature map $\varphi = Id$. This RKHS satisfies analogs of Assumption 1 and Assumption 2 in the sense that
> $$
>     \max \lbrace \sup_{\Vert z \Vert =1} \Vert v(z) \Vert , \ \sup_{\Vert z \Vert=1} \Vert Dv(z) \Vert, \ \sup_{\Vert z \Vert =1} \Vert D^2v(z) \Vert \rbrace \quad \leq \quad \Vert v \Vert_V ,
> $$
>
> and
> $$
>     \lambda_{\max}( \mathbb{K}((z^i))) = \sigma_{\max}(Z)^2, \quad \lambda_{\min}( \mathbb{K}((z^i))) = \sigma_{\min}(Z)^2,
> $$
>
> where $ \mathbb{K}((z^i))$ is the kernel matrix associated to the point cloud $(z^i)_i$ and $Z = (z^1 | ... | z^N)$.
>
> For a parameter $v \in L^2([0,1], V)$, define $U_1$ to be the solution at time $t=1$ of the linear ODE:
> $$
>     \frac{d}{d t} U_t  = v_t U_t , \quad U_0 = Id_q .
> $$
>
> The model of Definition 1 is then given by $F(v, x) = B U_1 A x$ and the loss of eq. (7) is $L(v) = \frac{1}{2N} \Vert B U_1 A X - Y \Vert_F^2$ where $X = (x^1 | ... | x^N)$, $Y = (y^1 | ... | y^N)$ are the data matrices. Also, because it is a linear regression problem, one can not always hope to to interpolate the data and we are therefore interested in minimizing the excess risk $\tilde{L}(v) = L(v) - L^*$ with $L^* = \min_{U \in  \mathbb{R}^{q \times q}} \frac{1}{2N} \Vert BUAX-Y \Vert_F^2$.
>
> Similarly to our Property 2, one can then show that the excess risk satisfies the following PL property:
> $$
>     2 m(\Vert v \Vert) \tilde{L}(v) \leq \Vert \nabla \tilde{L}(v) \Vert^2 \leq 2 M(\Vert v \Vert) \tilde{L}(v) .
> $$
>
> where, for every $R \geq 0$, $m(R)$ and $M(R)$ are given by:
> $$
>     m(R)  = \frac{1}{N} \sigma_{\min}(B^T)^2 \sigma_{\min}(A)^2 \sigma_r(X)^2 e^{-2R} , \quad
>     M(R)  = \frac{1}{N} \sigma_{\max}(B^T)^2 \sigma_{\max}(A)^2 \sigma_{\max}(X)^2 e^{-2R}
> $$
>
> and $\sigma_r(X)$ is the smallest non-zero singular value of $X$. Applying Theorem 1, one recovers convergence of GD as soon as there exists some $R > 0$ s.t.:
> $$
>     2 \sqrt{2} \frac{\sigma_{\max}(B^\top) \sigma_{\max}(A) \sigma_{\max}(X) \sqrt{L(0)-L^*}}{\sigma_{\min}(B^\top)^2 \sigma_{\min}(A)^2 \sigma_r(X)^2} \leq R e^{-3R} .
> $$
>
> This is achieved as soon as the l.h.s. is smaller than the universal constant $C = (3e)^{-1}$. This is the same convergence result as in "On the Global Convergence of training deep linear ResNets" (Zou et al. 2020), Theorem 3.1.

---

> ### Author Response · Authors · 2022-08-02
> **Can the analysis in line 256-258 be more concrete ? In there a principled way to construct such RKHS ?**
>
> We acknowledge that the hypotheses of Assumption 2 might look technical. They are basically satisfied by most reasonable universal kernels (such as a Gaussian of bandwidth $\sigma$), and $\Lambda$ and $\lambda$ (which will depends on the kernel, i.e. on $\sigma$ for instance) impacts the constants in the theorems.
>
> Assumption 2 is required in order to ensure the expressivity of our model. This expressivity is quantified by the conditioning of the kernel matrix and by $\Lambda$ and $\lambda$ which depends on the kernel. Therefore, the choice of kernel may have a significant impact on training. In theory, prior knowledge of the data distribution may allow to optimize the choice of kernel. However, in practice, we expect the selection of an optimal kernel to be an intractable problem. Instead, cross-validation techniques can be used to select a suitable kernel.

---

> ### Author Response · Authors · 2022-08-02
> **What's the relationship between your analysis and the Resnet's NTK ? What's the further degree of freedom increased in your paper ?**
>
> While the NTK also relies on a RKHS analysis, we believe our analysis is different and less local, since the NTK is dictated by the problem under study (in particular by the input data) while our RKHS is simply the one associated to the architecture of the deep network.
>
> More precisely, the model's NTK is defined for inputs $x, x'$ by:
> $$
> K_{NTK}(x, x') = \mathbb{E}_{v \sim \mathcal{P}} \langle \frac{\partial F}{\partial v}(v,x), \frac{\partial F}{\partial v}(v,x') \rangle
> $$
>
> where $\mathcal{P}$ is the network's weight distribution at initialization. In our model (Definition 1), the kernel $K$ is given by:
> $$
> K(x,x') = \langle \varphi(x), \varphi(x') \rangle
> $$
>
> where $\varphi$ is the feature map depending on the network architecture. For example, in the case where the residuals are 2-layer-MLPs (eq. (2)) with fixed hidden layer $U$ and non-linearity $\sigma$ this feature map is given by $\varphi : x \mapsto \sigma(Ux)$.
>
> We stress that there is no reason for the model's NTK and the kernel $K$ to be be the same.
>
> Also, and in sharp contrast with the NTK convergence analysis, our results still hold for finite width networks whereas NTK analysis is concerned with the infinite width limit.

---

> ### Author Response · Authors · 2022-08-02
> **Can the analysis in "On the Global Convergence of Gradient Descent for multi-layer ResNets in the mean-field regime" (Ding et al. 2021) be considered as a PL analysis ?**
>
> To our understanding no. We believe our work and the one in the reference provided by the reviewer to be of a different nature:
>
> - Our work relies on PL analysis : we show that a PL inequality is satisfied by our model in the case of universal kernel and that the condition for that PL inequality to be satisfied can be recovered in the case of finite-dimensional (i.e. parametric) kernels with high probability. The PL inequality then implies convergence of GD by classical arguments.
>
> - On the contrary, the work of Ding et al. studies ResNets in the mean-field (infinite width) regime, for which the training dynamic is expressed as a gradient flow in the space of probability measures endowed with the Wasserstein metric. In contrast with our work, they do not show a PL inequality is satisfied by the loss landscape for the Wasserstein metric. As a consequence, their result is a result of optimality at convergence and not a convergence result: Theorem 6.1 and 6.2 show that, **assuming convergence towards a limiting measure**, this measure is a minimizer of the loss.
>
> One of the reasons for this difference is that no PL inequality can be derived in full generality for mean-field models because of the existence of saddle points of the loss landscape for the Wasserstein metric: even if Proposition 6.1 shows that only global minimizers $\rho^*$ verify $\frac{\delta E}{\delta \rho}(\rho^*) = 0$, there exist measures (e.g. $\rho = 0$) such that $E(\rho) > 0$ and:
> $$
>     \Vert \nabla_W E(\rho) \Vert_{\rho}^2 = \Vert \nabla \frac{\delta E}{\delta \rho} (\rho) \Vert_{L^2(\rho)}^2 = \int \Vert \nabla \frac{\delta E}{\delta \rho} (\rho) \Vert^2 d \rho = 0
> $$
>
> where $\nabla_W$ is the gradient operator for the Wasserstein metric. Classical assumptions to circumvent this difficulty is for example homogeneity and full support of the initial measure on a sphere (e.g. separation condition in Theorem 6.1 and 6.2).

---

### Official Review · Reviewer_u7jP · 2022-07-08

**Rating:** 1
**Confidence:** 3
**Soundness:** 1 poor
**Presentation:** 1 poor
**Contribution:** 1 poor

**Summary:**

This paper violates the NeurIPS submission rules, by putting supplementary materials (page 14-31) into the main paper submission, and will not be reviewed.

**Questions:**

This paper violates the NeurIPS submission rules, by putting supplementary materials (page 14-31) into the main paper submission, and will not be reviewed.

**Limitations:**

This paper violates the NeurIPS submission rules, by putting supplementary materials (page 14-31) into the main paper submission, and will not be reviewed.

**Strengths And Weaknesses:**

This paper violates the NeurIPS submission rules, by putting supplementary materials (page 14-31) into the main paper submission, and will not be reviewed.

---

### Official Review · Reviewer_xZax · 2022-07-11

**Rating:** 6
**Confidence:** 4
**Soundness:** 3 good
**Presentation:** 4 excellent
**Contribution:** 3 good

**Summary:**

This paper studies the convergence of continuous-time residual networks with linear parameterization in the residual block trained by GD. Due to linear parameterization, the residual block functions are understood to be functions from an RKHS. Gradient of the loss function wrt the parameters (in this case a family of RKHS functions parameterized by time) is derived using standard techniques in optimal control. Then, a functional PL condition is proved for the gradient, which leads to local convergence. To achieve global convergence, two approaches---dimensionality lifting and random Fourier feature approximation---are studied for a special kernel. Final theorem says "global" convergence can be achieved when the network takes random Fourier features, is sufficiently wide, the matrices A and B are appropriately chosen, and the parameter is initialized from 0.

**Questions:**

1. To the reviewer's understanding, "global convergence" in optimization means the algorithm converges from any initialization, which does not need to be close to the minimum. However, in section 5 of the paper, convergence to global minimum is proven for a specific initialization, v^0=0. This is not global convergence. Is there a way to improve the results to real global convergence, i.e. the GD converges to global minimum from any (reasonable) initialization? If not the statements in the paper might need to be changed.

**Limitations:**

The limitations have been addressed in the paper.

**Strengths And Weaknesses:**

Originality: The novel parts of this work lie in the functional PL condition for the loss of the model studies, as well as the the approaches to extend local convergence to global convergence in section 5. Though, the reviewer has question regarding the "global convergence" which is left for the questions part. The continuous time residual networks with linear parameterization has been explored in previous works, e.g. in [1]. The local convergence results based on PL condition is also not novel to the current work.

Quality and clarity: The paper is clear well organized.

Significance: The analysis is theoretical significant because it reveals the good loss landscape for one type of flow-based model. The flow-based model is theoretically interesting because they are infinite-depth limit of deep neural networks, especially ResNets. It is also practically important, with neural ODE as a representative application.


[1] E W, Ma C, Wu L. Machine learning from a continuous viewpoint, I. Science China Mathematics. 2020 Nov;63(11):2233-66.

---

> ### Author Response · Authors · 2022-08-02
> **Is there a way to improve the results to real global convergence, i.e. the GD converges to global minimum from any (reasonable) initialization ?**
>
> We acknowledge that our use of "global" convergence can be misleading. In our meaning, it corresponds to convergence towards a global minimum for any input dataset, but the initialization may be constrained. We will clarify this and propose to modify the title with "On the global convergence of [..]" instead of "Global convergence of [..]". Nevertheless, we wanted to stress that this 0 initialization is a common practice for training deep ResNets: it was for example used in "Fixup initilization [...]" (Zhang et al. 2018) to efficiently train ResNets without normalization layers. When viewing training as an optimal control problem, 0 initialization means the map implemented by the network is initialized at identity.
>
> Finally, we prove in Proposition 4 (Appendix D.1) that the convergence result of Proposition 1 (and thereby Theorem 3) can be extended to the case of non-zero (but not too big) initializations. More precisely, Proposition 4 states convergence of GD for large enough $q$ when considering initializations whose norms decay as $q^{-1/4}$:
> $$
>         \Vert v^0_q \Vert_{V_q} = \mathcal{O}(q^{-1/4}) .
> $$
>
> This technical assumption is required in order for the model's output at initialization not to blow when increasing the dimension $q$ (see line 897 in SM). Concretely, when considering the residual space in eq. (3) with the feature map of eq. (20) (Random Fourier Features) and the weight matrix $W$ initialized with i.i.d. Gaussian entries the assumption holds when adding a $q^{-3/4}$ scaling factor. For example, the result holds when considering the residual space:
> $$
>         V_q = \lbrace v : z \mapsto W \varphi(z), \ W \in \mathbb{R}^{q \times r} \rbrace , \quad \text{with} \quad \varphi(z) = r^{-1/2} ( e^{\imath \langle z, \omega^j \rangle} )_{1 \leq j \leq r}
> $$
>
> and random initialization $W_{i,j}^0 \sim \mathcal{N}(0,q^{-3/2}) $.
>
> We think that it might be possible that global convergence is achieved with high-probability for random initialization of the outer weights of the residual blocks. It is left as an open question.

---

> > ### Comment · Reviewer_xZax · 2022-08-08
> > **Thank for the response**
> >
> > The response has addressed my question and clarified the meaning of "global convergence" in the paper. The reviewer will keep the score.

---

### Author Response · Authors · 2022-08-08
**Final updates and comments**

Dear reviewers,

We highly appreciated your comments and would like to take the last opportunity to address your follow-up concerns, as the discussion will be closed tomorrow.

Regarding the concern of reviewer u7jP, we believe it is acceptable to have the supplementary material in the same PDF as the paper, and this is not a cause of rejection.

Best regards,

---

### Comment · Reviewer_xZax · 2022-08-08
**About the "violation rules"**

The reviewer will respect any decision by the AC, but hope this paper will not get rejected only because the careless review by u7jP. One point I want to raise is that the review u7jP chose "3" in the confidence part, while he/she obvious did not read the paper at all. This shows unprofessional and hence this review should be disregarded.

---

> ### Comment · Reviewer_u7jP · 2022-08-08
> **Ad hominem**
>
> The submission rules should be quite clear from the CFP.
> https://neurips.cc/Conferences/2022/CallForPapers
>
> In particular,
> 1. "Submissions are limited to nine content pages"
> 2. "Submissions that violate the NeurIPS style ... or page limits may be rejected without further review."
> 3. "Papers may be rejected without consideration of their merits if they fail to meet the submission requirements"
>
> Reviewer xZax just made an ad hominem fallacy.

---

> > ### Comment · Reviewer_jSRi · 2022-08-08
> > **It's a rude review**
> >
> > The paper have nine content page, all the other stuff are in the appendix
> > This can be simply fixed by split the pdf. It's very rude to reject a paper by this means.
> >
> > I hope the ac can ignore this rude reviewer.

---

> > > ### Author Response · Authors · 2022-08-08
> > > **Thanks a lot for your support**
> > >
> > > We would like to thank you for the update and your support.

---

> ### Author Response · Authors · 2022-08-08
> **Thank you for your support**
>
> We would like to thank you for the update and also for your support. We agree with you that it should be the role of the AC to decide wether or not the paper should be desk-rejected.

---

### Meta-Review · Area_Chair_LkxQ · 2022-08-25

**Recommendation:** Accept
**Confidence:** Certain

**Metareview:**

The paper presents convergence analysis for ResNet in a certain asymptotic regime, for which the authors are able to establish local Polyak-Lojasiewicz inequality. The analysis sheds new light on the convergence of neural network training. It is a technical sound paper and merits acceptance to the conference.

**Award:**

No

---

### Decision · Program_Chairs · 2022-09-14

Accept